# VMamba: Visual State Space Model

**Yue Liu [1]**    **Yunjie Tian[1]**    **Yuzhong Zhao[1]**    **Hongtian Yu[1]**
**Lingxi Xie[2]**    **Yaowei Wang[3]**    **Qixiang Ye[1]**    **Jianbin Jiao[1]**    **Yunfan Liu[1‡]**
[1] UCAS    [2] Huawei Inc.    [3] Pengcheng Lab.
{liuyue171,tianyunjie19,zhaoyuzhong20,yuhongtian17}@mails.ucas.ac.cn
198808xc@gmail.com, wangyw@pcl.ac.cn, {qxye,jiaojb,liuyunfan}@ucas.ac.cn

## Abstract

Designing computationally efficient network architectures remains an ongoing necessity in computer vision. In this paper, we adapt Mamba, a state-space language model, into VMamba, a vision backbone with linear time complexity. At the core of VMamba is a stack of Visual State-Space (VSS) blocks with the 2D Selective Scan (SS2D) module. By traversing along four scanning routes, SS2D bridges the gap between the ordered nature of 1D selective scan and the non-sequential structure of 2D vision data, which facilitates the collection of contextual information from various sources and perspectives. Based on the VSS blocks, we develop a family of VMamba architectures and accelerate them through a succession of architectural and implementation enhancements. Extensive experiments demonstrate VMamba's promising performance across diverse visual perception tasks, highlighting its superior input scaling efficiency compared to existing benchmark models. Source code is available at `https://github.com/MzeroMiko/VMamba`

## 1 Introduction

Visual representation learning remains as a fundamental research area in computer vision that has witnessed remarkable progress in the era of deep learning. To represent complex patterns in vision data, two primary categories of backbone networks, *i.e.*, Convolutional Neural Networks (CNNs) [49, 27, 29, 53, 37] and Vision Transformers (ViTs) [13, 36, 57, 66], have been proposed and extensively utilized in a variety of visual tasks. Compared to CNNs, ViTs generally demonstrate superior learning capabilities on large-scale data due to their integration of the self-attention mechanism [58, 13]. However, the quadratic complexity of self-attention w.r.t. the number of tokens imposes substantial computational overhead in downstream tasks involving large spatial resolutions.

To address this challenge, significant efforts have been made to improve the efficiency of attention computation [54, 36, 12]. However, existing approaches either restrict the size of the effective receptive field [36] or suffer from notable performance degradation across various tasks [30, 60]. This motivates us to develop a novel architecture for vision data, while maintaining the inherent advantages of the vanilla self-attention mechanism, *i.e.*, global receptive fields and dynamic weighting parameters [23].

Recently, Mamba [17], a innovative State Space Model (SSM) [17, 43, 59, 71, 48], in the field of natural language processing (NLP), has emerged as a promising approach for long-sequence modeling with linear complexity. Drawing inspiration from this advancement, we introduce VMamba, a vision backbone that integrates SSM-based blocks to enable efficient visual representation learning. However, the core algorithm of Mamba, *i.e.*, the parallelized selective scan operation, is essentially designed for processing one-dimensional sequential data. This presents a challenge when adapting it for processing vision data, which lacks an inherent sequential arrangement of visual components. To address this issue, we propose 2D Selective Scan (SS2D), a four-way scanning mechanism designed for spatial domain traversal. In contrast to the self-attention mechanism (Figure 1 (a)), SS2D ensures

38th Conference on Neural Information Processing Systems (NeurIPS 2024).

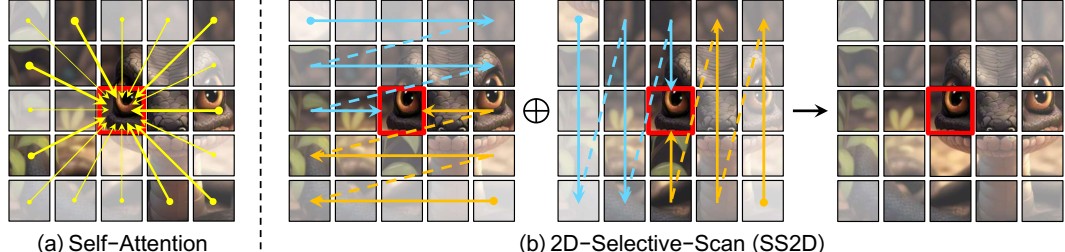

(a) Self−Attention          (b) 2D−Selective−Scan (SS2D)

Figure 1: Comparison of the establishment of correlations between image patches through (a) self-attention and (b) the proposed 2D-Selective-Scan (SS2D). The red boxes indicate the query image patch, with its opacity representing the degree of information loss.

that each image patch acquires contextual knowledge exclusively through a compressed hidden state computed along its corresponding scanning path (Figure 1 (b)), thereby reducing the computational complexity from quadratic to linear.

Building on the VSS blocks, we develop a family of VMamba architectures (*i.e.*, VMamba-Tiny/Small/Base) and enhance their performance through architectural improvements and implementation optimizations. Compared to benchmark vision models built on CNNs (ConvNeXt [37]), ViTs (Swin [36], HiViT [66]), and SSMs (S4ND [44], Vim [69]), VMamba consistently achieves higher image classification accuracy on ImageNet-1K [9] across various model scales. Specifically, VMamba-Base achieves a top-1 accuracy of $83.9\%$, surpassing Swin by $+0.4\%$, with a throughput exceeding Swin's by a substantial margin over $40\%$ ($646$ *vs.* $458$). VMamba's superiority extends across multiple downstream tasks, with VMamba-Tiny/Small/Base achieving $47.3\%/48.7\%/49.2\%$ mAP in object detection on COCO [33] ($1\times$ training schedule). This outperforms Swin by $4.6\%/3.9\%/2.3\%$ and ConvNeXt by $3.1\%/3.3\%/2.2\%$, respectively. As for single-scale semantic segmentation on ADE20K [68], VMamba-Tiny/Small/Base achieves $47.9\%/50.6\%/51.0\%$ mIoU, which surpasses Swin by $3.4\%/3.0\%/2.9\%$ and ConvNeXt by $1.9\%/1.9\%/1.9\%$, respectively. Furthermore, unlike ViT-based models, which experience quadratic growth in computational complexity with the number of input tokens, VMamba exhibits linear growth in FLOPs while maintaining comparable performance. This demonstrates its state-of-the-art input scalability.

The contributions of this study are summarized as follows:

- We propose VMamba, an SSM-based vision backbone for visual representation learning with linear time complexity. A series of architectural and implementation improvements are adopted to enhance the inference speed of VMamba.
- We introduce 2D Selective Scan (SS2D) to bridge 1D array scanning and 2D plane traversal, enabling the extension of selective SSMs to process vision data.
- VMamba achieves promising performance across various visual tasks, including image classification, object detection, and semantic segmentation. It also exhibits remarkable adaptability w.r.t. the length of the input sequence, showcasing linear growth in computational complexity.

## 2 Related Work

**Convolutional Neural Networks (CNNs).** Since AlexNet [31], considerable efforts have been devoted to enhancing the modeling capabilities [49, 52, 27, 29] and computational efficiency [28, 53, 64, 46] of CNN-based models across various visual tasks. Sophisticated operators like depth-wise convolution [28] and deformable convolution [5, 70] have been introduced to increase the flexibility and efficacy of CNNs. Recently, inspired by the success of Transformers [58], modern CNNs [37] have shown promising performance by integrating long-range dependencies [11, 47, 34] and dynamic weights [23] into their designs.

**Vision Transformers (ViTs).** As a pioneering work, ViT [13] explores the effectiveness of vision models based on vanilla Transformer architecture, highlighting the importance of large-scale

pre-training for image classification performance. To reduce ViT's dependence on large datasets, DeiT [57] introduces a teacher-student distillation strategy, transferring knowledge from CNNs to ViTs and emphasizing the importance of inductive bias in visual perception. Following this approach, subsequent studies propose hierarchical ViTs [36, 12, 61, 39, 66, 55, 6, 10, 67, 1].

Another research direction focuses on improving the computational efficiency of self-attention, which serves as the cornerstone of ViTs. Linear Attention [30] reformulates self-attention as a linear dot-product of kernel feature maps, using the associativity property of matrix products to reduce computational complexity from quadratic to linear. GLA [65] introduces a hardware-efficient variant of linear attention that balances memory movement with parallelizability. RWKV [45] also leverages the linear attention mechanism to combine parallelizable transformer training with the efficient inference of recurrent neural networks (RNNs). RetNet [51] adds a gating mechanism to enable a parallelizable computation path, offering an alternative to recurrence. RMT [15] further extends this for visual representation learning by applying the temporal decay mechanism to the spatial domain.

**State Space Models (SSMs).**    Despite their widespread adoption in vision tasks, ViT architectures face significant challenges due to the quadratic complexity of self-attention, especially when handling long input sequences (*e.g.*, high-resolution images). In efforts to improve scaling efficiency [8, 7, 45, 51, 41], SSMs have emerged as compelling alternatives to Transformers, attracting significant research attention. Gu *et al.* [21] demonstrate the potential of SSM-based models in handling the long-range dependencies using the HiPPO initialization [18]. To improve practical feasibility, S4 [20] proposes normalizing the parameter matrices into a diagonal structure. Various structured SSM models have since emerged, each offering distinct architectural enhancements, such as complex-diagonal structures [22, 19], support for multiple-input multiple-output [50], diagonal plus low-rank decomposition [24], and selection mechanisms [17]. These advancements have also been integrated into larger representation models [43, 41, 16], further highlighting the versatility and scalability of structured state space models in various applications. While these models primarily target long-range and sequential data such as text and speech, limited research has explored applying SSMs to vision data with two-dimensional structures.

## 3    Preliminaries

**Formulation of SSMs.**    Originating from the Kalman filter [32], SSMs are linear time-invariant (LTI) systems that map the input signal $u(t) \in \mathbb{R}$ to the output response $y(t) \in \mathbb{R}$ via the hidden state $\mathbf{h}(t) \in \mathbb{R}^N$. Specifically, continuous-time SSMs can be expressed as linear ordinary differential equations (ODEs) as follows,

$$\begin{aligned} \mathbf{h}'(t) &= \mathbf{A}\mathbf{h}(t) + \mathbf{B}u(t), \\ y(t) &= \mathbf{C}\mathbf{h}(t) + Du(t), \end{aligned} \tag{1}$$

where $\mathbf{A} \in \mathbb{R}^{N \times N}$, $\mathbf{B} \in \mathbb{R}^{N \times 1}$, $\mathbf{C} \in \mathbb{R}^{1 \times N}$, and $D \in \mathbb{R}^1$ are the weighting parameters.

**Discretization of SSM.**    To be integrated into deep models, continuous-time SSMs must undergo discretization in advance. Concretely, for the time interval $[t_a, t_b]$, the analytic solution of the hidden state variable $\mathbf{h}(t)$ at $t = t_b$ can be expressed as

$$\mathbf{h}(t_b) = e^{\mathbf{A}(t_b - t_a)}\mathbf{h}(t_a) + e^{\mathbf{A}(t_b - t_a)} \int_{t_a}^{t_b} \mathbf{B}(\tau)u(\tau)e^{-\mathbf{A}(\tau - t_a)} \, d\tau. \tag{2}$$

By sampling with the time-scale parameter $\mathbf{\Delta}$ (*i.e.*, $d\tau|_{t_i}^{t_{i+1}} = \Delta_i$), $h(t_b)$ can be discretized by

$$\mathbf{h}_b = e^{\mathbf{A}(\Delta_a + \dots + \Delta_{b-1})} \left( \mathbf{h}_a + \sum_{i=a}^{b-1} \mathbf{B}_i u_i e^{-\mathbf{A}(\Delta_a + \dots + \Delta_i)} \Delta_i \right), \tag{3}$$

where $[a, b]$ is the corresponding discrete step interval. Notably, this formulation approximates the result obtained by the zero-order hold (ZOH) method, which is frequently utilized in the literature of SSM-based models (please refer to Appendix A for detailed proof).

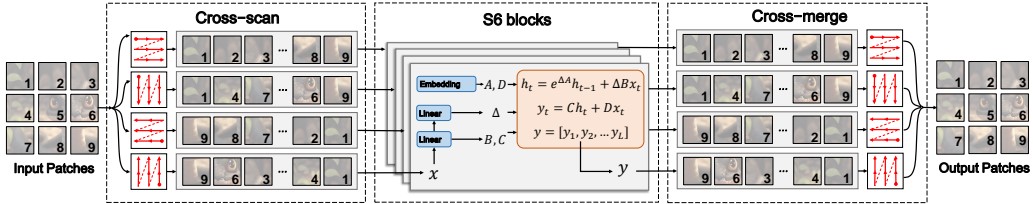

Figure 2: Illustration of 2D-Selective-Scan (SS2D). Input patches are traversed along four different scanning paths (*Cross-Scan*), with each sequence independently processed by separate S6 blocks. The results are then merged to construct a 2D feature map as the final output (*Cross-Merge*).

**Selective Scan Mechanism.** To address the limitation of LTI SSMs (Eq. 1) in capturing the contextual information, Gu *et al.* [17] propose a novel parameterization method for SSMs, which incorporates an input-dependent selection mechanism (referred to as S6). However, for selective SSMs, the time-varying weighting parameters pose a challenge for efficient computation of hidden states, as convolutions cannot accommodate dynamic weights, making them inapplicable. Nevertheless, since the recurrence relation of $h_b$ in Eq. 3 can be derived, the response $y_b$ can still be efficiently computed using associative scan algorithms [2, 42, 50], which has linear complexity (see Appendix B for a detailed explanation).

## 4 VMamba: Visual State Space Model

### 4.1 Network Architecture

We develop VMamba in three scales: Tiny, Small, and Base (referred to as VMamba-T, VMamba-S, and VMamba-B, respectively). An overview of the architecture of VMamba-T is illustrated in Figure 3 (a), and detailed configurations are provided in Appendix E. The input image $\mathbf{I} \in \mathbb{R}^{H \times W \times 3}$ is first partitioned into patches by a stem module, resulting in a 2D feature map with spatial dimension of $H/4 \times W/4$. Without incorporating additional positional embeddings, multiple network stages are employed to create hierarchical representations with resolutions of $H/8 \times W/8$, $H/16 \times W/16$, and $H/32 \times W/32$. Specifically, each stage comprises a down-sampling layer (except for the first stage), followed by a stack of Visual State Space (VSS) blocks.

The VSS blocks serve as the visual counterparts to Mamba blocks [17] (Figure 3 (b)) for representation learning. The initial architecture of VSS blocks (referred to as the 'vanilla VSS Block' in Figure 3 (c)) is formulated by replacing the S6 module. S6 is the core of Mamba and achieves global receptive fields, dynamic weights (*i.e.*, selectivity), and linear complexity. We substitute it with the newly proposed 2D-Selective-Scan (SS2D) module, and more details will be introduced in the following subsection. To further enhance computational efficiency, we remove the entire multiplicative branch (highlighted by the red box in Figure 3 (c)), as the effect of the gating mechanism has already been achieved by the selectivity of SS2D. As a result, the improved VSS block (shown in Figure 3 (d)) consists of a single network branch with two residual modules, mimicking the architecture of a vanilla Transformer block [58]. All results in this paper are obtained using VMamba models built with VSS blocks in this architecture.

### 4.2 2D-Selective-Scan for Vision Data (SS2D)

While the sequential nature of the scanning operation in S6 aligns well with NLP tasks involving temporal data, it poses a significant challenge when applied to vision data, which is inherently non-sequential and encompasses spatial information (*e.g.*, local texture and global structure). To address this issue, S4ND [44] reformulates SSM with convolutional operations, directly extending the kernel from 1D to 2D through the outer-product. However, such modification restricts the weights from being input-dependent, resulting in a limited capacity for capturing contextual information. Therefore, we adhere to the selective scan approach [17] for input processing and propose the 2D-Selective-Scan (SS2D) module to adapt S6 to vision data without compromising its advantages.

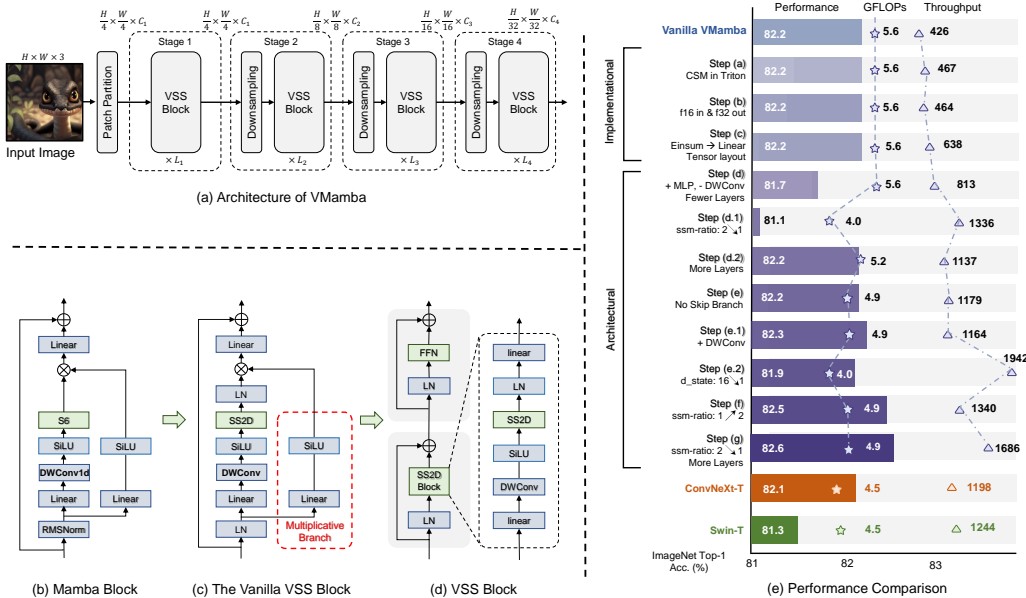

Figure 3: **Left:** Illustration of (a) the overall architecture of VMamba, and (b) - (d) the structure of Mamba and VSS blocks. **Right:** Comparison of VMamba variants and benchmark methods in terms of classification accuracy and computational efficiency.

Figure 2 illustrates that data forwarding in SS2D consists of three steps: cross-scan, selective scanning with S6 blocks, and cross-merge. Specifically, SS2D first unfolds the input patches into sequences along four distinct traversal paths (*i.e.*, Cross-Scan). Each patch sequence is then processed in parallel using a separate S6 block, and the resultant sequences are reshaped and merged to form the output map (*i.e.*, Cross-Merge). Through the use of complementary 1D traversal paths, SS2D allows each pixel in the image to integrate information from all other pixels across different directions. This integration facilitates the establishment of global receptive fields in the 2D space.

### 4.3 Accelerating VMamba

As shown in Figure 3 (e), the VMamba-T model with vanilla VSS blocks (referred to as 'Vanilla VMamba') achieves a throughput of 426 images/s and contains 22.9M parameters with 5.6G FLOPs. Despite achieving a state-of-the-art classification accuracy of 82.2% (outperforming Swin-T [36] by 0.9% at the tiny level), the low throughput and high memory overhead present significant challenges for the practical deployment of VMamba.

In this subsection, we outline our efforts to enhance its inference speed, primarily focusing on improvements in both implementation details and architectural design. We evaluate the models with image classification on ImageNet-1K. The impact of each progressive improvement is summarized as follows, where (%, img/s) denote the gains in top-1 accuracy on ImageNet-1K and inference throughput, respectively. Further discussion is provided in Appendix E.

Step (a) $(+0.0\%, +41$ img/s) by re-implementing Cross-Scan and Cross-Merge in `Triton`.

Step (b) $(+0.0\%, -3$ img/s) by adjusting the CUDA implementation of selective scan to accommodate `float16` input and `float32` output. This remarkably enhances the training efficiency (throughput from 165 to 184), despite slight speed fluctuation at test time.

Step (c) $(+0.0\%, +174$ img/s) by substituting the relatively slow `einsum` in selective scan with a linear transformation (*i.e.*, `torch.nn.functional.linear`). We also adopt the tensor layout of (`B, C, H, W`) to eliminate unnecessary data permutations.

Table 1: Performance comparison on ImageNet-1K. Throughput values are measured with an A100 GPU and an AMD EPYC 7542 CPU, using the toolkit released by [62], following the protocol proposed in [36]. All images are of size $224 \times 224$.

| Model | Params (M) | FLOPs (G) | TP. (img/s) | Top-1 (%) | Model | Params (M) | FLOPs (G) | TP. (img/s) | Top-1 (%) |
|---|---|---|---|---|---|---|---|---|---|
| **Transformer-Based** | | | | | **ConvNet-Based** | | | | |
| DeiT-S [57] | 22M | 4.6G | 1761 | 79.8 | ConvNeXt-T [37] | 29M | 4.5G | 1198 | 82.1 |
| DeiT-B [57] | 86M | 17.5G | 503 | 81.8 | ConvNeXt-S [37] | 50M | 8.7G | 684 | 83.1 |
| HiViT-T [66] | 19M | 4.6G | 1393 | 82.1 | ConvNeXt-B [37] | 89M | 15.4G | 436 | 83.8 |
| HiViT-S [66] | 38M | 9.1G | 712 | 83.5 | **SSM-Based** | | | | |
| HiViT-B [66] | 66M | 15.9G | 456 | 83.8 | S4ND-Conv-T [44] | 30M | 5.2G | 683 | 82.2 |
| Swin-T [36] | 28M | 4.5G | 1244 | 81.3 | S4ND-ViT-B [44] | 89M | 17.1G | 397 | 80.4 |
| Swin-S [36] | 50M | 8.7G | 718 | 83.0 | Vim-S [69] | 26M | 5.3G | 811 | 80.5 |
| Swin-B [36] | 88M | 15.4G | 458 | 83.5 | VMamba-T | 30M | 4.9G | 1686 | 82.6 |
| XCiT-S24 [1] | 48M | 9.2G | 671 | 82.6 | VMamba-S | 50M | 8.7G | 877 | 83.6 |
| XCiT-M24 [1] | 84M | 16.2G | 423 | 82.7 | VMamba-B | 89M | 15.4G | 646 | 83.9 |

Step (d) $(-0.6\%, +175$ img/s$)$ by introducing `MLP` into VMamba due to its computational efficiency. We also discard the `DWConv` (depth-wise convolutional [23]) layers and change the layer configuration from `[2,2,9,2]` to `[2,2,2,2]` to lower FLOPs.

Step (e) $(+0.6\%, +366$ img/s$)$ by reducing the parameter `ssm-ratio` (the feature expansion factor) from 2.0 to 1.0 (also referred to as Step (d.1)), raising the layer numbers to `[2,2,5,2]` (also referred to as Step (d.2)), and discarding the entire multiplicative branch as illustrated in Figure 3 (c).

Step (f) $(+0.3\%, +161$ img/s$)$ by introducing the `DWConv` layers (also referred to as Step (e.1)) and reducing the parameter `d_state` (the SSM state dimension) from 16.0 to 1.0 (also referred to as Step (e.2)), together with raising `ssm-ratio` back to 2.0.

Step (g) $(+0.1\%, +346$ img/s$)$ by reducing the `ssm-ratio` to 1.0 while changing the layer configuration from `[2,2,5,2]` to `[2,2,8,2]`.

# 5    Experiments

In this section, we present a series of experiments to evaluate the performance of VMamba and compare it to popular benchmark models across various visual tasks. We also validate the effectiveness of the proposed 2D feature map traversal method by comparing it with alternative approaches. Additionally, we analyze the characteristics of VMamba by visualizing its effective receptive field (ERF) and activation map, and examining its scalability with longer input sequences. We primarily follow the hyperparameter settings and experimental configurations used in Swin [36]. For detailed experiment settings, please refer to Appendix E and F, and for additional ablations, see Appendix H. All experiments were conducted on a server with $8 \times$ NVIDIA Tesla-A100 GPUs.

## 5.1    Image Classification

We evaluate VMamba's performance in image classification on ImageNet-1K [9], with comparison results against benchmark methods summarized in Table 1. With similar FLOPs, VMamba-T achieves a top-1 accuracy of $82.6\%$, outperforming DeiT-S by $2.8\%$ and Swin-T by $1.3\%$. Notably, VMamba maintains its performance advantage at both Small and Base scales. For example, VMamba-B achieves a top-1 accuracy of $83.9\%$, surpassing DeiT-B by $2.1\%$ and Swin-B by $0.4\%$.

In terms of computational efficiency, VMamba-T achieves a throughput of 1,686 images/s, which is either superior or comparable to state-of-the-art methods. This advantage continues with VMamba-S and VMamba-B, achieving throughputs of 877 images/s and 646 images/s, respectively. Compared to SSM-based models, the throughput of VMamba-T is $1.47\times$ higher than S4ND-Conv-T [44] and $1.08\times$ higher than Vim-S [69], while maintaining a clear performance lead of $0.4\%$ and $2.1\%$ over these models, respectively.

Table 2: **Left:** Results for object detection and instance segmentation on MSCOCO. $AP^b$ and $AP^m$ denote box AP and mask AP, respectively. FLOPs are calculated with an input size of $1280 \times 800$. The notation '1×' indicates models fine-tuned for 12 epochs, while '3×MS' denotes multi-scale training for 36 epochs. **Right:** Results for semantic segmentation on ADE20K. FLOPs are calculated with an input size of $512 \times 2048$. 'SS' and 'MS' denote single-scale and multi-scale testing, respectively.

| Mask R-CNN 1× schedule | | | | |
|---|---|---|---|---|
| Backbone | $AP^b$ | $AP^m$ | Params | FLOPs |
| Swin-T | 42.7 | 39.3 | 48M | 267G |
| ConvNeXt-T | 44.2 | 40.1 | 48M | 262G |
| VMamba-T | 47.3 | 42.7 | 50M | 271G |
| Swin-S | 44.8 | 40.9 | 69M | 354G |
| ConvNeXt-S | 45.4 | 41.8 | 70M | 348G |
| VMamba-S | 48.7 | 43.7 | 70M | 349G |
| Swin-B | 46.9 | 42.3 | 107M | 496G |
| ConvNeXt-B | 47.0 | 42.7 | 108M | 486G |
| VMamba-B | 49.2 | 44.1 | 108M | 485G |
| Mask R-CNN 3× MS schedule | | | | |
| Swin-T | 46.0 | 41.6 | 48M | 267G |
| ConvNeXt-T | 46.2 | 41.7 | 48M | 262G |
| NAT-T | 47.7 | 42.6 | 48M | 258G |
| VMamba-T | 48.8 | 43.7 | 50M | 271G |
| Swin-S | 48.2 | 43.2 | 69M | 354G |
| ConvNeXt-S | 47.9 | 42.9 | 70M | 348G |
| NAT-S | 48.4 | 43.2 | 70M | 330G |
| VMamba-S | 49.9 | 44.2 | 70M | 349G |

| ADE20K with crop size 512 | | | | |
|---|---|---|---|---|
| Backbone | mIOU (SS) | mIOU (MS) | Params | FLOPs |
| ResNet-50 | 42.1 | 42.8 | 67M | 953G |
| DeiT-S + MLN | 43.8 | 45.1 | 58M | 1217G |
| Swin-T | 44.5 | 45.8 | 60M | 945G |
| ConvNeXt-T | 46.0 | 46.7 | 60M | 939G |
| NAT-T | 47.1 | 48.4 | 58M | 934G |
| Vim-S | 44.9 | - | 46M | - |
| VMamba-T | 47.9 | 48.8 | 62M | 949G |
| ResNet-101 | 43.8 | 44.9 | 86M | 1030G |
| DeiT-B + MLN | 45.5 | 47.2 | 144M | 2007G |
| Swin-S | 47.6 | 49.5 | 81M | 1039G |
| ConvNeXt-S | 48.7 | 49.6 | 82M | 1027G |
| NAT-S | 48.0 | 49.5 | 82M | 1010G |
| VMamba-S | 50.6 | 51.2 | 82M | 1028G |
| Swin-B | 48.1 | 49.7 | 121M | 1188G |
| ConvNeXt-B | 49.1 | 49.9 | 122M | 1170G |
| NAT-B | 48.5 | 49.7 | 123M | 1137G |
| RepLKNet-31B | 49.9 | 50.6 | 112M | 1170G |
| VMamba-B | 51.0 | 51.6 | 122M | 1170G |

## 5.2 Downstream Tasks

In this sub-section, we evaluate the performance of VMamba on downstream tasks, including object detection and instance segmentation on MSCOCO2017 [33], and semantic segmentation on ADE20K [68]. The training framework is based on the MMDetection [3] and MMSegmenation [4] libraries, following [35] in utilizing Mask R-CNN [26] and UperNet [63] as the detection and segmentation networks, respectively.

**Object Detection and Instance Segmentation.** The results on MSCOCO are presented in Table 2. VMamba demonstrates superior performance in both box and mask Average Precision ($AP^b$ and $AP^m$) across different training schedules. Under the 12-epoch fine-tuning schedule, VMamba-T/S/B achieves object detection mAPs of $47.3\%/48.7\%/49.2\%$, outperforming Swin-T/S/B by $4.6\%/3.9\%/2.3\%$ mAP and ConvNeXt-T/S/B by $3.1\%/3.3\%/2.2\%$ mAP, respectively. VMamba-T/S/B achieves instance segmentation mAPs that exceed Swin-T/S/B by $3.4\%/2.8\%/1.8\%$ mAP and ConvNeXt-T/S/B by $2.6\%/1.9\%/1.4\%$ mAP, respectively. Furthermore, VMamba's advantages persist with the 36-epoch fine-tuning schedule using multi-scale training, highlighting its strong potential in downstream tasks requiring dense predictions.

**Semantic Segmentation.** Consistent with previous experiments, VMamba demonstrates superior performance in semantic segmentation on ADE20K with a comparable amount of parameters. As shown in Table 2, VMamba-T achieves $3.4\%$ higher mIoU than Swin-T and $1.9\%$ higher than ConvNeXt-T in the Single-Scale (SS) setting, and the advantage persists with Multi-Scale (MS) input. For models at the Small and Base levels, VMamba-S/B outperforms NAT-S/B [25] by $2.6\%/2.5\%$ mIoU in the SS setting, and $1.7\%/1.9\%$ mIoU in the MS setting.

**Discussion** The experimental results in this subsection demonstrate VMamba's adaptability to object detection, instance segmentation, and semantic segmentation. In Figure 4 (a), we compare VMamba's performance with Swin and ConvNeXt, highlighting its advantages in handling downstream tasks with comparable classification accuracy on ImageNet-1K. This result aligns with Figure 4 (b), where VMamba shows the most stable performance (*i.e.*, modest performance drop) across different input image sizes, achieving a top-1 classification accuracy of $74.7\%$ without fine-tuning ($79.2\%$ with `linear tuning`) at an input resolution of $768 \times 768$. While exhibiting greater tolerance to changes

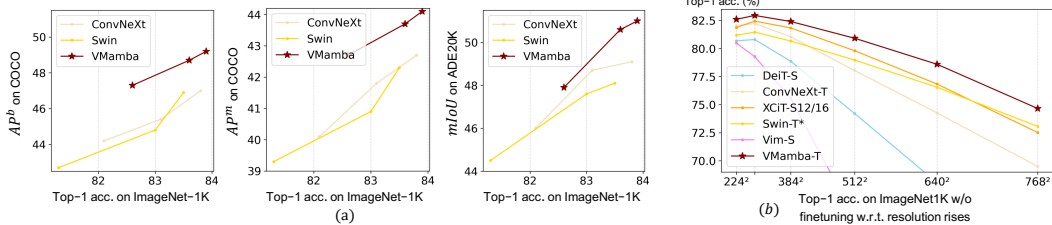

Figure 4: Illustration of VMamba's adaptability to (a) downstream tasks and (b) input images with progressively increasing resolutions. Swin-T* denotes Swin-T tested with scaled window sizes.

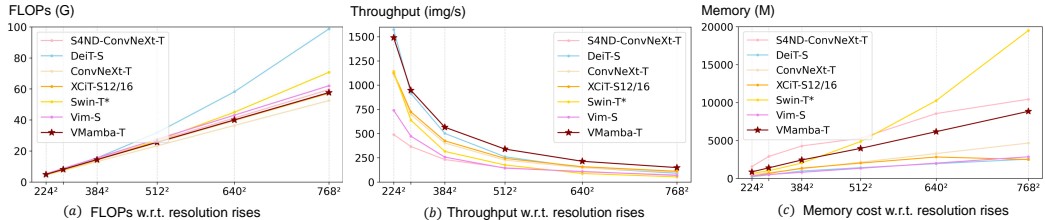

Figure 5: Illustration of VMamba's resource consumption with progressively increasing resolutions. Swin-T* denotes Swin-T tested with scaled window sizes.

in input resolution, VMamba also maintains linear growth in FLOPs and memory-consumption (see Figure 5 (a) and (c)) and maintains high throughput ( Figure 5 (b)), making it more effective and efficient compared to ViT-based methods when adapting to downstream tasks with inputs of larger spatial resolutions. This aligns with Mamba's advanced capability in efficient long sequence modeling [17].

## 5.3 Analysis

**Relationship between SS2D and Self-Attention.** To formulate the response $\mathbf{Y}$ within the time interval $[a, b]$ of length $T$, we denote the corresponding SSM-related variables $\mathbf{u}_i \odot \boldsymbol{\Delta}_i \in \mathbb{R}^{1 \times D_v}$, $\mathbf{B}_i \in \mathbb{R}^{1 \times D_k}$, and $\mathbf{C}_i \in \mathbb{R}^{1 \times D_k}$ as $\mathbf{V} \in \mathbb{R}^{T \times D_v}$, $\mathbf{K} \in \mathbb{R}^{T \times D_k}$, and $\mathbf{Q} \in \mathbb{R}^{T \times D_k}$, respectively. Therefore, the $j$-th slice along dimension $D_v$ of $\mathbf{y_b}$, denoted as $\mathbf{y_b}^{(j)} \in \mathbb{R}$ can be written as

$$\mathbf{y_b}^{(j)} = \left( \mathbf{Q_T} \odot \mathbf{w_T}^{(j)} \right) \mathbf{h_a}^{(j)} + \mathbf{Q_T} \sum_{i=1}^{T} \left( \frac{\mathbf{w_T}^{(j)}}{\mathbf{w_i}^{(j)}} \odot \mathbf{K_i} \right)^{\top} \odot \left( \mathbf{V_i}^{(j)} \right). \quad (4)$$

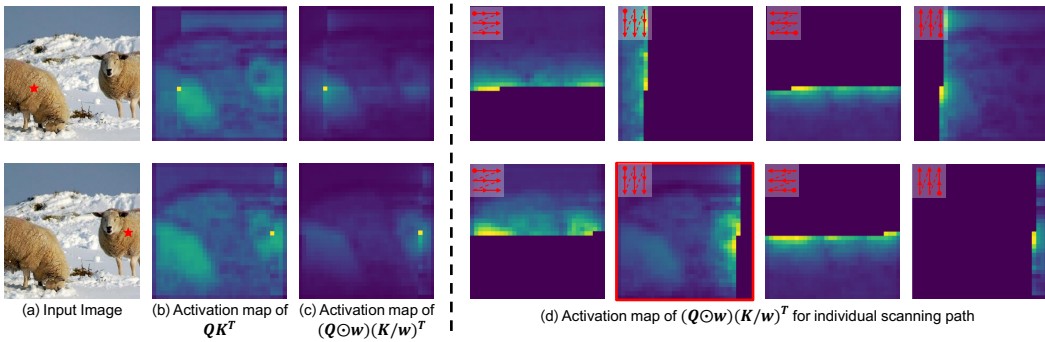

Figure 6: Illustration of the activation map for query patches indicated by red stars. The visualization results in (b) and (c) are obtained by combining the activation maps from each scanning path in SS2D.

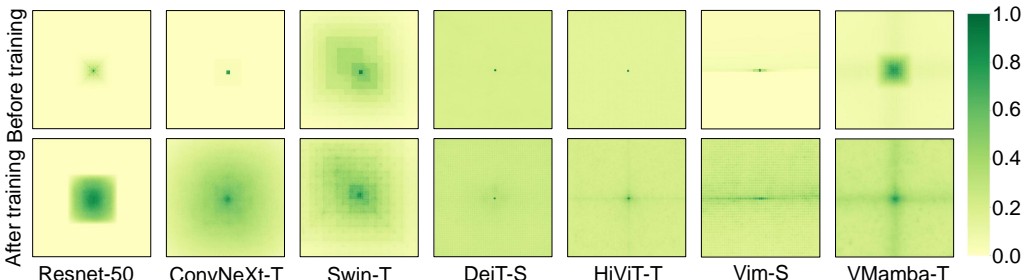

Figure 7: Comparison of Effective Receptive Fields (ERF) [40] between VMamba and other benchmark models. Pixels with higher intensity indicate larger responses related to the central pixel.

where $\mathbf{h_a} \in \mathbb{R}^{D_k}$ is the hidden state at step $a$, $\odot$ denotes element-wise product. Particularly, $\mathbf{V_i}^{(j)}$ is only a scalar. The formulation of each element in $\mathbf{w} := [\mathbf{w}_1; \dots ; \mathbf{w}_T] \in \mathbb{R}^{T \times D_k \times D_v}$, $i.e.$, $\mathbf{w}_i \in \mathbb{R}^{D_k \times D_v}$, can be written as $\mathbf{w}_i = \prod_{j=1}^{i} e^{\mathbf{A}\mathbf{\Delta}_{a-1+j}^\top}$, representing the cumulative attention weight at step $i$ computed along the scanning path.

Consequently, the $j$-th dimension of $\mathbf{Y}$, $i.e.$, $\mathbf{Y}^{(j)} \in \mathbb{R}^{T \times 1}$, can be expressed as

$$\mathbf{Y}^{(j)} = \left[\mathbf{Q} \odot \mathbf{w}^{(j)}\right]\mathbf{h_a}^{(j)} + \left[\left(\mathbf{Q} \odot \mathbf{w}^{(j)}\right)\left(\frac{\mathbf{K}}{\mathbf{w}^{(j)}}\right)^\top \odot \mathbf{M}\right]\mathbf{V}^{(j)}, \tag{5}$$

where $\mathbf{M}$ denotes the temporal mask matrix of size $T \times T$ with the lower triangular part set to 1 and elsewhere 0. Please refer to Appendix C for more detailed derivations.

In Eq. 5, the matrix multiplication process involving $\mathbf{Q}$, $\mathbf{K}$, and $\mathbf{V}$ closely resembles the self-attention mechanism, despite the inclusion of $\mathbf{w}$.

**Visualization of Activation Maps.** To gain an intuitive and in-depth understanding of SS2D, we further visualize the attention values in $\mathbf{Q}\mathbf{K}^\top$ and $(\mathbf{Q} \odot \mathbf{w})(\mathbf{K}/\mathbf{w})^\top$ corresponding to a specific query patch within foreground objects (referred to as the 'activation map'). As shown in Figure 6 (b), the activation map of $\mathbf{Q}\mathbf{K}^\top$ demonstrates the effectiveness of SS2D in capturing and retaining traversed information, with all previously scanned tokens in the foreground region being activated. Furthermore, the inclusion of $\mathbf{w}$ results in activation maps that are more focused on the neighborhood of query patches (Figure 6 (c)), which is consistent with the temporal weighting effect inherent in the formulation of $\mathbf{w}$. Nevertheless, the selective scan mechanism allows VMamba to accumulate history along the scanning path, facilitating the establishment of long-term dependencies across image patches. This is evident in the sub-figure encircled by a red box (Figure 6 (d)), where patches of the sheep far to the left (scanned in earlier steps) remain activated. For more visualizations and further discussion, please refer to Appendix D.

**Visualization of Effective Receptive Fields.** The Effective Receptive Field (ERF) [40, 11] refers to the region in the input space that contributes to the activation of a specific output unit. We conduct a comparative analysis of the central pixel's ERF across various visual backbones, both before and after training. The results presented in Figure 7 illustrate that among the models examined, only DeiT, HiViT, Vim and VMamba demonstrate global ERFs, while the others exhibit local ERFs despite their theoretical potential for global coverage. Moreover, VMamba's linear time complexity enhances its computational efficiency compared to DeiT and HiViT, which incur quadratic costs w.r.t. the number of input patches. While both VMamba and Vim are based on the Mamba architecture, VMamba's ERF is more uniform and 2D-aware than that of Vim, which may intuitively explain its superior performance.

**Diagnostic Study on Selective Scan Patterns.** We compare the proposed scanning pattern (*i.e.* Cross-Scan) to three benchmark patterns: unidirectional scanning (Unidi-Scan), bidirectional scanning (Bidi-Scan), and cascade scanning (Cascade-Scan, scanning the data row-wise and column-wise successively). Feature dimensions are adjusted to maintain similar architectural parameters and

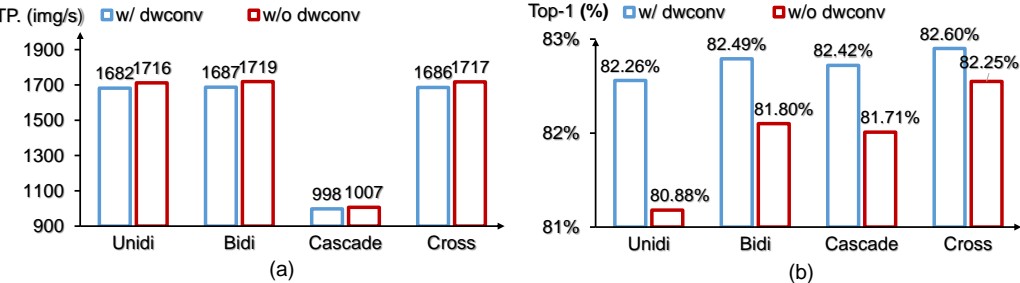

Figure 8: Performance comparison of different scanning patterns. The proposed Cross-Scan achieves superior performance in speed while maintaining the same number of parameters and FLOPs.

FLOPs for a fair comparison. As illustrated in Figure 8, Cross-Scan outperforms the other scanning patterns in both computational efficiency and classification accuracy, highlighting its effectiveness in achieving 2D-Selective-Scan. Removing the DWConv layer, which has been shown to aid the model in learning 2D spatial information, further enhances this advantage. This underscores the inherent strength of Cross-Scan in capturing 2D contextual information through its adoption of four-way scanning.

## 6 Conclusion

This paper presents VMamba, an efficient vision backbone model built with State Space Models (SSMs). VMamba integrates the advantages of selective SSMs from NLP tasks into visual data processing, bridging the gap between ordered 1D scanning and non-sequential 2D traversal through the novel SS2D module. Furthermore, we have significantly improved the inference speed of VMamba through a series of architectural and implementation refinements. The effectiveness of the VMamba family has been demonstrated through extensive experiments, and its linear time complexity makes VMamba advantageous for downstream tasks with large-resolution inputs.

**Limitations.** While VMamba demonstrates promising experimental results, there is still room for improvement in this study. Previous research has validated the efficacy of unsupervised pre-training on large-scale datasets (*e.g.*, ImageNet-21K). However, the compatibility of existing pre-training methods with SSM-based architectures like VMamba, as well as the identification of pre-training techniques specifically tailored for such models, remain unexplored. Investigating these aspects could serve as a promising avenue for future research in architectural design. Additionally, limited computational resources have prevented us from exploring VMamba's architecture at the Large scale and conducting a fine-grained hyperparameter search to further enhance experimental performance. Although SS2D, the core component of VMamba, does not make specific assumptions about the layout or modality of the input data, allowing it to generalize across various tasks, the potential of VMamba for integration into more generalized tasks remains unexplored. Bridging the gap between SS2D and these tasks, along with proposing a more generalized scanning pattern for vision tasks, represents a promising research direction.

## 7 Acknowledgments

This work was supported by National Natural Science Foundation of China (NSFC) under Grant No.62225208 and 62406304, CAS Project for Young Scientists in Basic Research under Grant No.YSBR-117, China Postdoctoral Science Foundation under Grant No.2023M743442, and Postdoctoral Fellowship Program of CPSF under Grant No.GZB20240730.

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

## A  Discretization of State Space Models (SSMs)

In this section, we explore the correlation between the discretized formulations of State Space Models (SSMs) obtained in Sec. 3 and those derived from the zero-order hold (ZOH) method [17], which is frequently used in studies related to SSMs.

Recall the discretized formulation of SSMs derived in Sec. 3 as follows,

$$\mathbf{h}_b = e^{\mathbf{A}(\Delta_a + \ldots + \Delta_{b-1})} \left( \mathbf{h}_a + \sum_{i=a}^{b-1} \mathbf{B}_i u_i e^{-\mathbf{A}(\Delta_a + \ldots + \Delta_i)} \Delta_i \right). \tag{6}$$

Let $b = a + 1$, then the above equation can be re-written as

$$\mathbf{h}_{a+1} = e^{\mathbf{A}\Delta_a} \mathbf{h}_a + \mathbf{B}_a \Delta_a u_a, \tag{7}$$

where $\overline{\mathbf{A_a}} := e^{\mathbf{A}\Delta_a}$ is the exact discretized form of the evolution matrix $\mathbf{A}$ obtained by ZOH, and $\overline{\mathbf{B_a}} := \mathbf{B}_a \Delta_a$ represents the first-order Taylor expansion of the discretized $\mathbf{B}$ acquired through ZOH.

## B  Derivation of the Recurrence Relation of Selective SSMs

In this section, we derive the recurrence relation of the hidden state in selective SSMs. Given the expression of $h_b$ shown in Eq. 6, let us denote $e^{\mathbf{A}(\Delta_a + \ldots + \Delta_{i-1})}$ as $\mathbf{p}_{\mathbf{A,a}}^{\mathbf{i}}$. Then, its recurrence relation can be directly written as

$$\mathbf{p}_{\mathbf{A,a}}^{\mathbf{i}} = e^{\mathbf{A}\Delta_{i-1}} \mathbf{p}_{\mathbf{A,a}}^{\mathbf{i-1}}. \tag{8}$$

For the second term of Eq. 6, we have

$$\mathbf{p}_{\mathbf{B,a}}^{\mathbf{b}} = e^{\mathbf{A}(\Delta_a + \ldots + \Delta_{b-1})} \sum_{i=a}^{b-1} \mathbf{B_i} u_i e^{-\mathbf{A}(\Delta_a + \ldots + \Delta_i)} \Delta_i \tag{9}$$

$$= e^{\mathbf{A}\Delta_{b-1}} \mathbf{p}_{\mathbf{B,a}}^{\mathbf{b-1}} + \mathbf{B_{b-1}} u_{b-1} \Delta_{b-1}. \tag{10}$$

Therefore, with the associations derived in Eq. 8 and Eq. 9, $\mathbf{h}_b = \mathbf{p}_{\mathbf{A,a}}^{\mathbf{b}} \mathbf{h}_a + \mathbf{p}_{\mathbf{B,a}}^{\mathbf{b}}$ can be efficiently computed in parallel using associative scan algorithms [2, 42, 50], which are supported by numerous modern programming libraries. This approach effectively reduces the overall computational complexity to linear, and VMamba further accelerates the computation by adopting a hardware-aware implementation [17].

## C  Details of the relationship between SS2D and Self-attention

In this section, we clarify the relationship between SS2D and the self-attention mechanism commonly employed in existing vision backbone models. Subsequently, visualization results are provided to substantiate our explanation.

Let $T$ denote the length of the sequence with indices from $a$ to $b$, we define the following variables

$$\mathbf{V} := [\mathbf{V_1}; \ldots; \mathbf{V_T}] \in \mathbb{R}^{T \times D_v}, \text{ where } \mathbf{V_i} := \mathbf{u_{a+i-1}} \odot \mathbf{\Delta_{a+i-1}} \in \mathbb{R}^{1 \times D_v} \tag{11}$$

$$\mathbf{K} := [\mathbf{K_1}; \ldots; \mathbf{K_T}] \in \mathbb{R}^{T \times D_k}, \text{ where } \mathbf{K_i} := \mathbf{B_{a+i-1}} \in \mathbb{R}^{1 \times D_k} \tag{12}$$

$$\mathbf{Q} := [\mathbf{Q_1}; \ldots; \mathbf{Q_T}] \in \mathbb{R}^{T \times D_k}, \text{ where } \mathbf{Q_i} := \mathbf{C_{a+i-1}} \in \mathbb{R}^{1 \times D_k} \tag{13}$$

$$\mathbf{w} := [\mathbf{w_1}; \ldots; \mathbf{w_T}] \in \mathbb{R}^{T \times D_k \times D_v}, \text{ where } \mathbf{w_i} := \prod_{j=1}^{i} e^{\mathbf{A}\mathbf{\Delta}_{a-1+j}^{\top}} \in \mathbb{R}^{D_k \times D_v} \tag{14}$$

$$\mathbf{H} := [\mathbf{h_{a+1}}; \ldots; \mathbf{h_b}] \in \mathbb{R}^{T \times D_k \times D_v}, \text{ where } \mathbf{h_i} \in \mathbb{R}^{D_k \times D_v} \tag{15}$$

$$\mathbf{Y} := [\mathbf{y_{a+1}}; \ldots; \mathbf{y_b}] \in \mathbb{R}^{T \times D_v}, \text{ where } \mathbf{y_i} \in \mathbb{R}^{D_v} \tag{16}$$

Note that in practice, the parameter $A$ in Eq. 1 is simplified to $\mathbb{R}^{1 \times D_k}$. Consequently, $\mathbf{h}'(t) = \mathbf{A}\mathbf{h}(t) + \mathbf{B}u(t)$ is simplified to $\mathbf{h}'(t) = \mathbf{A} \odot \mathbf{h}(t) + \mathbf{B}u(t)$, which is the reason why $\mathbf{w_i} \in \mathbb{R}^{D_k \times D_v}$.

Based on these notations, the discretized solution of time-varying SSMs (Eq. 6) can be written as

$$\mathbf{h_b} = \mathbf{w_T} \odot \mathbf{h_a} + \sum_{i=1}^{T} \frac{\mathbf{w_T}}{\mathbf{w_i}} \odot \left( \mathbf{K_i}^\top \mathbf{V_i} \right), \tag{17}$$

where $\odot$ denotes the element-wise product between matrices, and the division is also elements-wise.

Based on the expression of the hidden state $\mathbf{h_b}$, the first term of the output of SSM, *i.e.*, $\mathbf{y_b}$, can be computed by

$$\mathbf{y_b} = \mathbf{Q_T} \mathbf{h_b} \tag{18}$$

$$= \mathbf{Q_T} \left( \mathbf{w_T} \odot \mathbf{h_a} \right) + \mathbf{Q_T} \sum_{i=1}^{T} \frac{\mathbf{w_T}}{\mathbf{w_i}} \odot \left( \mathbf{K_i}^\top \mathbf{V_i} \right). \tag{19}$$

Here, we drop the skip connection between the input and the response for simplicity. Particularly, the $j$-th slice along dimension $D_v$ of $\mathbf{y_b}$, denoted as $\mathbf{y_b}^{(j)} \in \mathbb{R}$ can be written as

$$\mathbf{y_b}^{(j)} = \left( \mathbf{Q_T} \odot \mathbf{w_T}^{(j)} \right) \mathbf{h_a}^{(j)} + \sum_{i=1}^{T} \left( \frac{\mathbf{Q_T} \odot \mathbf{w_T}^{(j)}}{\mathbf{w_i}^{(j)}} \mathbf{K_i}^\top \right) \odot \mathbf{V_i}^{(j)}. \tag{20}$$

Similarly, the $j$-th slice along dimension $D_v$ of the overall response $\mathbf{Y}$, denoted as $\mathbf{Y}^{(j)} \in \mathbb{R}^{T \times 1}$, can be expressed as

$$\mathbf{Y}^{(j)} = \left( \mathbf{Q} \odot \mathbf{w}^{(j)} \right) \mathbf{h_a}^{(j)} + \left[ \left( \mathbf{Q} \odot \mathbf{w}^{(j)} \right) \left( \frac{\mathbf{K}}{\mathbf{w}^{(j)}} \right)^\top \odot \mathbf{M} \right] \mathbf{V}^{(j)}, \tag{21}$$

where $\mathbf{M} := \texttt{tril}(T, T) \in \{0, 1\}^{T \times T}$ denotes the temporal mask matrix with the lower triangular portion of a $T \times T$ matrix set to 1 and elsewhere 0. It is evident that how matrices $\mathbf{Q}$, $\mathbf{K}$, and $\mathbf{V}$ are multiplied in Eq. 21 closely resembles the process in the self-attention module of Vision Transformers. Moreover, if $\mathbf{w}$ is in shape $(T, D_k)$ rather than $(T, D_k, D_v)$, then Eq. 18 and Eq. 21 reduce to the form of Gated Linear Attention (GLA) [65], indicating that GLA is also a special case of Mamba.

## D   Visualization of Attention and Activation Maps

In the preceding subsection, we illustrated how the computational process of selective SSMs shares similarities with self-attention mechanisms, allowing us to delve into the internal mechanism of SS2D through the visualization of its weight matrices.

Given the input image shown in Figure 9 (a), illustrations of four scanning paths in SS2D are presented in Figure 9 (d). The visualizations of the corresponding attention maps, calculated using $\mathbf{Q}\mathbf{K}^\top$ and $(\mathbf{Q} \odot \mathbf{w}) (\mathbf{K}/\mathbf{w})^\top$ are shown in Figure 9 (e) and Figure 9 (g) respectively. These results underscore the effectiveness of the proposed scanning approach (*i.e.*, Cross-Scan) in capturing and retaining the traversed information, as each row in a single attention map corresponds to the attention between the current patch and all previously scanned foreground tokens impartially. Additionally, in Figure 9 (f), we showcase the transformed activation maps, where the pixel order corresponds to that of the first route, traversing the image row-wise from the upper-left to the bottom-right.

By rearranging the diagonal elements of the obtained attention map in the image space, we derive the visualization results shown in Figure 9 (b) and Figure 9 (c) corresponding to $\mathbf{Q}\mathbf{K}^\top$ and $(\mathbf{Q} \odot \mathbf{w}) (\mathbf{K}/\mathbf{w})^\top$ respectively. These maps illustrate the effectiveness of VMamba in accurately distinguishing between foreground and background pixels within an image.

Moreover, given a selected patch as the query, we visualize the corresponding activation map by reshaping the associated row in the attention map (computed by $\mathbf{Q}\mathbf{K}^\top$ or $(\mathbf{Q} \odot \mathbf{w}) (\mathbf{K}/\mathbf{w})^\top$) This reflects the attention score between the query patch and all previously scanned patches. To obtain the complete visualization for a query patch, we collect and combine the activation maps from all four scanning paths in SS2D. The visualization results of the activation map for both $\mathbf{Q}\mathbf{K}^\top$ and $(\mathbf{Q} \odot \mathbf{w}) (\mathbf{K}/\mathbf{w})^\top$ are shown in Figure 10. We also visualize the diagonal elements of attention maps computed by $(\mathbf{Q} \odot \mathbf{w}) (\mathbf{K}/\mathbf{w})^\top$, where all foreground objects are effectively highlighted and separated from the background.

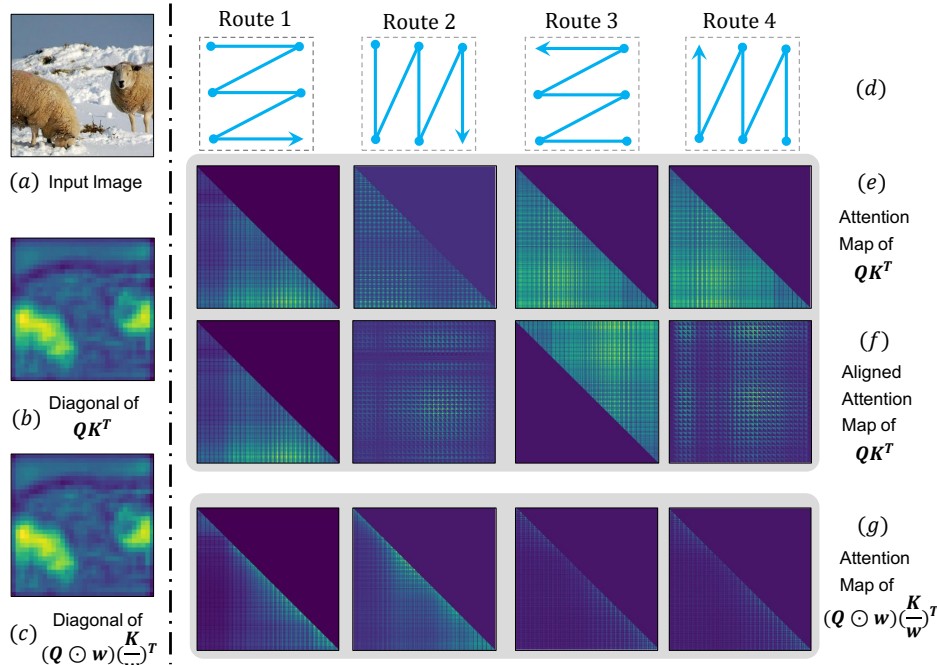

Figure 9: Illustration of the attention maps obtained by SS2D.

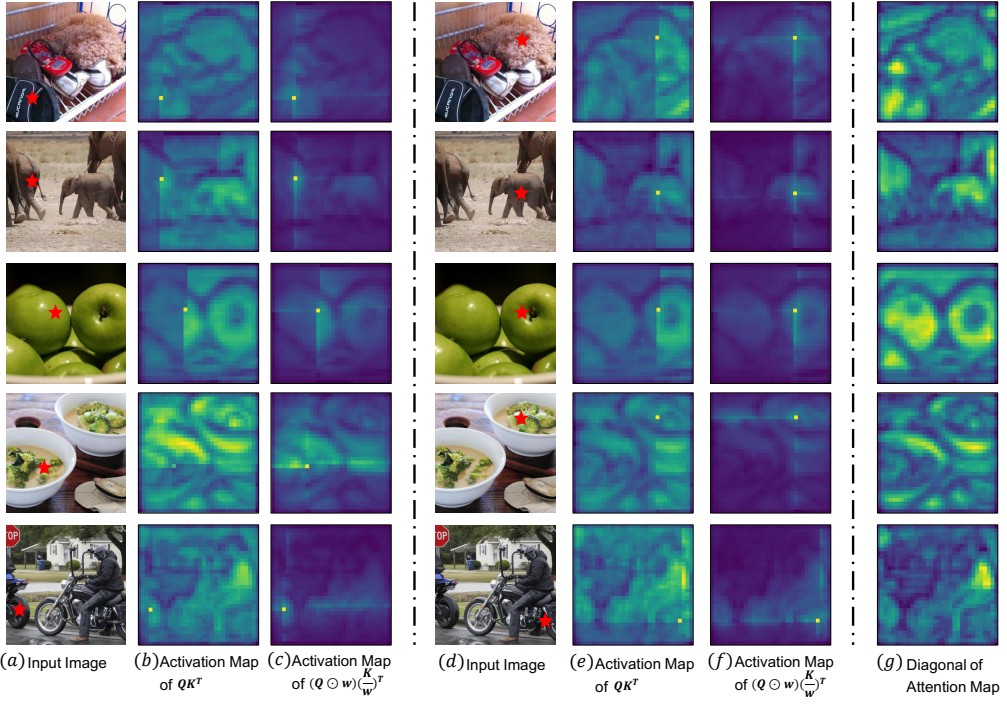

Figure 10: Illustration of activation maps for the query patch (marked with a red star).

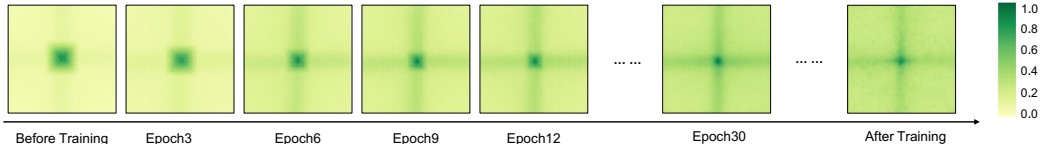

Figure 11: Illustration of ERF maps throughout the training process of Vanilla-VMamba-T (with EMA).

# E   Detailed Experiment Settings

**Network Architecture.**    The architectural specifications of Vanilla-VMamba are outlined in Table 3, while detailed configurations of the VMamba series are provided in Table 4. The Vanilla-VMamba series is constructed using the vanilla VSS Block, which includes a multiplicative branch and does not have feed-forward network (FFN) layers. In contrast, the VSS Block in the VMamba series removes the multiplicative branch and introduces FFN layers. Additionally, we provide alternative architectures for VMamba at Small and Base scales, referred to as VMamba-S[$s1l20$] and VMamba-B[$s1l20$], respectively. The notation '$sxly$' indicates that the `ssm-ratio` is set to $x$ and the number of layers in stage 3 is set to $y$. Consequently, the versions presented in Table 1 can also be referred to as VMamba-S[$s2l15$] and VMamba-B[$s2l15$].

**Experiment Setting.**    The hyper-parameters for training VMamba on ImageNet are inherited from Swin [36], except for the parameters related to `drop_path_rate` and the exponential moving average (EMA) technique. Specifically, VMamba-T/S/B models are trained from scratch for 300 epochs, with a 20-epoch warm-up period, using a batch size of 1024. The training process utilizes the AdamW optimizer [38] with betas set to $(0.9, 0.999)$, an initial learning rate of $1 \times 10^{-3}$, a weight decay of 0.05, and a cosine decay learning rate scheduler. It is noteworthy that this is not the optimal setting for VMamba. With a learning rate of $2 \times 10^{-3}$, the Top-1 accuracy of VMamba-T can reach 80.7%.

Additional techniques such as label smoothing (0.1) and EMA (decay ratio of 0.9999) are also applied. The `drop_path_ratio` is set to 0.2 for Vanilla-VMamba-T and VMamba-T, 0.3 for Vanilla-VMamba-S, VMamba-S[$s2l15$] and VMamba-S[$s1l20$], 0.6 for Vanilla-VMamba-B and VMamba-B[$s2l15$], and 0.5 for VMamba-B[$s1l20$]. No additional training techniques are employed.

**Throughput Evaluation.**    Detailed performance comparisons with various models are presented in Table 6. Throughput (referred to as `TP.`) was assessed on an A100 GPU paired with an AMD EPYC 7542 CPU, utilizing the toolkit provided by [62]. Following the protocol outlined in [36], we set the batch size to 128. The training throughput (referred to as `Train TP.`) is tested on the same device with `mix-resolution`, excluding the time consumption of optimizers. The batch size for measuring the training throughput is also set to 128.

**Accelerating VMamba.**    Table 5 provides detailed configurations of the intermediate variants in the acceleration process from Vanilla-VMamba-T to VMamba-T.

**Evolution of ERF.**    We further generate the effective receptive field (ERF) maps throughout the training process for Vanilla-VMamba-T. These maps intuitively illustrate how VMamba's pattern of ERF evolves from being predominantly local to predominantly global, epoch by epoch.

# F   Performance of the VMamba Family on Downstream Tasks

In this section, we present the experimental results of Vanilla-VMamba and VMamba on the MSCOCO and ADE20k datasets. The results are summarized in Table 7 and Table 8, respectively.

For object detection and instance segmentation, we adhere to the protocol outlined by Swin [36] and construct our models using the mmdetection framework [3]. Specifically, we utilize the AdamW optimizer [38] and fine-tune the classification models pre-trained on ImageNet-1K for both 12 and 36 epochs. The learning rate is initialized at $1 \times 10^{-4}$ and decreased by a factor of 10 at the 9-*th*

Table 3: Architectural overview of the Vanilla-VMamba series. Down-sampling is performed through patch merging [36] operations in stages 1, 2, and 3. The term `Linear` refers to a linear layer, while `DWConv` denotes a depth-wise convolution [23] operation. The proposed 2D-selective-scan is labeled as SS2D.

| layer name | output size | Vanilla-VMamba-T | Vanilla-VMamba-S | Vanilla-VMamba-B |
|---|---|---|---|---|
| stem | 112×112 | conv 4×4, 96, stride 4 | conv 4×4, 96, stride 4 | conv 4×4, 128, stride 4 |
| stage 1 | 56×56 | vanilla VSSBLock $\begin{bmatrix} \text{Linear } 96 \to 2\times96 \\ \text{DWConv } 3\times3,\ 2\times96 \\ \text{SS2D, dim } 2\times96 \\ \text{Linear } 2\times96 \to 96 \\ \text{Multiplicative} \\ \text{Linear } 2\times96 \to 96 \end{bmatrix} \times2$ | vanilla VSSBLock $\begin{bmatrix} \text{Linear } 96 \to 2\times96 \\ \text{DWConv } 3\times3,\ 2\times96 \\ \text{SS2D, dim } 2\times96 \\ \text{Linear } 2\times96 \to 96 \\ \text{Multiplicative} \\ \text{Linear } 2\times96 \to 96 \end{bmatrix} \times2$ | vanilla VSSBLock $\begin{bmatrix} \text{Linear } 128 \to 2\times128 \\ \text{DWConv } 3\times3,\ 2\times128 \\ \text{SS2D, dim } 2\times128 \\ \text{Linear } 2\times128 \to 128 \\ \text{Multiplicative} \\ \text{Linear } 2\times128 \to 128 \end{bmatrix} \times2$ |
| | | patch merging → 192 | patch merging → 192 | patch merging → 256 |
| stage 2 | 28×28 | vanilla VSSBLock $\begin{bmatrix} \text{Linear } 192 \to 2\times192 \\ \text{DWConv } 3\times3,\ 2\times192 \\ \text{SS2D, dim } 2\times192 \\ \text{Linear } 2\times192 \to 192 \\ \text{Multiplicative} \\ \text{Linear } 2\times192 \to 192 \end{bmatrix} \times2$ | vanilla VSSBLock $\begin{bmatrix} \text{Linear } 192 \to 2\times192 \\ \text{DWConv } 3\times3,\ 2\times192 \\ \text{SS2D, dim } 2\times192 \\ \text{Linear } 2\times192 \to 192 \\ \text{Multiplicative} \\ \text{Linear } 2\times192 \to 192 \end{bmatrix} \times2$ | vanilla VSSBLock $\begin{bmatrix} \text{Linear } 256 \to 2\times256 \\ \text{DWConv } 3\times3,\ 2\times256 \\ \text{SS2D, dim } 2\times256 \\ \text{Linear } 2\times256 \to 256 \\ \text{Multiplicative} \\ \text{Linear } 2\times256 \to 256 \end{bmatrix} \times2$ |
| | | patch merging → 384 | patch merging → 384 | patch merging → 512 |
| stage 3 | 14×14 | vanilla VSSBLock $\begin{bmatrix} \text{Linear } 384 \to 2\times384 \\ \text{DWConv } 3\times3,\ 2\times384 \\ \text{SS2D, dim } 2\times384 \\ \text{Linear } 2\times384 \to 384 \\ \text{Multiplicative} \\ \text{Linear } 2\times384 \to 384 \end{bmatrix} \times9$ | vanilla VSSBLock $\begin{bmatrix} \text{Linear } 384 \to 2\times384 \\ \text{DWConv } 3\times3,\ 2\times384 \\ \text{SS2D, dim } 2\times384 \\ \text{Linear } 2\times384 \to 384 \\ \text{Multiplicative} \\ \text{Linear } 2\times384 \to 384 \end{bmatrix} \times27$ | vanilla VSSBLock $\begin{bmatrix} \text{Linear } 512 \to 2\times512 \\ \text{DWConv } 3\times3,\ 2\times512 \\ \text{SS2D, dim } 2\times512 \\ \text{Linear } 2\times512 \to 512 \\ \text{Multiplicative} \\ \text{Linear } 2\times512 \to 512 \end{bmatrix} \times27$ |
| | | patch merging → 768 | patch merging → 768 | patch merging → 1024 |
| stage 4 | 7×7 | vanilla VSSBLock $\begin{bmatrix} \text{Linear } 768 \to 2\times768 \\ \text{DWConv } 3\times3,\ 2\times768 \\ \text{SS2D, dim } 2\times768 \\ \text{Linear } 2\times768 \to 768 \\ \text{Multiplicative} \\ \text{Linear } 2\times768 \to 768 \end{bmatrix} \times2$ | vanilla VSSBLock $\begin{bmatrix} \text{Linear } 768 \to 2\times768 \\ \text{DWConv } 3\times3,\ 2\times768 \\ \text{SS2D, dim } 2\times768 \\ \text{Linear } 2\times768 \to 768 \\ \text{Multiplicative} \\ \text{Linear } 2\times768 \to 768 \end{bmatrix} \times2$ | vanilla VSSBLock $\begin{bmatrix} \text{Linear } 1024 \to 2\times1024 \\ \text{DWConv } 3\times3,\ 2\times1024 \\ \text{SS2D, dim } 2\times1024 \\ \text{Linear } 2\times1024 \to 1024 \\ \text{Multiplicative} \\ \text{Linear } 2\times1024 \to 1024 \end{bmatrix} \times2$ |
| | 1×1 | average pool, 1000-d fc, softmax | | |
| Param. (M) | | 22.9 | 44.4 | 76.3 |
| FLOPs | | $5.63\times10^9$ | $11.23\times10^9$ | $18.02\times10^9$ |

and 11-*th* epoch. We incorporate multi-scale training and random flipping with a batch size of 16, following established practices for object detection evaluations.

For semantic segmentation, we follow Swin [36] and construct a UperHead [63] network on top of the pre-trained model using the MMSegmentation library [4]. We employ the AdamW optimizer [38] and set the learning rate to $6 \times 10^{-5}$. The fine-tuning process spans a total of $160k$ iterations with a batch size of 16. The default input resolution is $512 \times 512$.

## G   Details of VMamba's Scale-Up Experiments

Given Mamba's exceptional ability in efficient long sequence modeling, we conduct experiments to assess whether VMamba inherits this characteristic. We evaluate the computational efficiency and classification accuracy of VMamba with progressively larger input spatial resolutions. Specifically, following the protocol in XCiT [1], we apply VMamba, trained on $224 \times 224$ inputs, to images with resolutions ranging from $288 \times 288$ to $768 \times 768$. We measure the generalization performance in terms of the number of parameters, FLOPs, throughput during both training and inference, and

Table 4: Architectural overview of the VMamba series.

| layer name | output size | VMamba-T | VMamba-S | VMamba-B |
|---|---|---|---|---|
| stem | 112×112 | conv 3×3 stride 2, LayerNorm, GeLU, conv 3×3 stride 2, LayerNorm | | |
| stage 1 | 56×56 | VSSBLock(ssm-ratio=1, mlp-ratio=4)
[ Linear 96 → ssm-ratio ×96
DWConv 3×3, ssm-ratio ×96
SS2D, dim ssm-ratio ×96
Linear ssm-ratio ×96 → 96
FFN mlp-ratio ×96 ] ×2 | VSSBlock(ssm-ratio=2, mlp-ratio=4)
[ Linear 96 → ssm-ratio ×96
DWConv 3×3, ssm-ratio ×96
SS2D, dim ssm-ratio ×96
Linear ssm-ratio ×96 → 96
FFN mlp-ratio ×96 ] ×2 | VSSBLock(ssm-ratio=2, mlp-ratio=4)
[ Linear 128 → ssm-ratio ×128
DWConv 3×3, ssm-ratio ×128
SS2D, dim ssm-ratio ×128
Linear ssm-ratio ×128 → 128
FFN mlp-ratio ×128 ] ×2 |
| | | conv 3×3 stride 2, LayerNorm | | |
| stage 2 | 28×28 | VSSBLock(ssm-ratio=1, mlp-ratio=4)
[ Linear 192 → ssm-ratio ×192
DWConv 3×3, ssm-ratio ×192
SS2D, dim ssm-ratio ×192
Linear ssm-ratio ×192 → 192
FFN mlp-ratio ×192 ] ×2 | VSSBlock(ssm-ratio=2, mlp-ratio=4)
[ Linear 96 → ssm-ratio ×192
DWConv 3×3, ssm-ratio ×192
SS2D, dim ssm-ratio ×192
Linear ssm-ratio ×192 → 192
FFN mlp-ratio ×192 ] ×2 | VSSBLock(ssm-ratio=2, mlp-ratio=4)
[ Linear 256 → ssm-ratio ×256
DWConv 3×3, ssm-ratio ×256
SS2D, dim ssm-ratio ×256
Linear ssm-ratio ×256 → 256
FFN mlp-ratio ×256 ] ×2 |
| | | conv 3×3 stride 2, LayerNorm | | |
| stage 3 | 14×14 | VSSBLock(ssm-ratio=1, mlp-ratio=4)
[ Linear 384 → ssm-ratio ×384
DWConv 3×3, ssm-ratio ×384
SS2D, dim ssm-ratio ×384
Linear ssm-ratio ×384 → 384
FFN mlp-ratio ×384 ] ×8 | VSSBlock(ssm-ratio=2, mlp-ratio=4)
[ Linear 384 → ssm-ratio ×384
DWConv 3×3, ssm-ratio ×384
SS2D, dim ssm-ratio ×384
Linear ssm-ratio ×384 → 384
FFN mlp-ratio ×384 ] ×15 | VSSBLock(ssm-ratio=2, mlp-ratio=4)
[ Linear 512 → ssm-ratio ×512
DWConv 3×3, ssm-ratio ×512
SS2D, dim ssm-ratio ×512
Linear ssm-ratio ×512 → 512
FFN mlp-ratio ×512 ] ×15 |
| | | conv 3×3 stride 2, LayerNorm | | |
| stage 4 | 7×7 | VSSBLock(ssm-ratio=1, mlp-ratio=4)
[ Linear 768 → ssm-ratio ×768
DWConv 3×3, ssm-ratio ×768
SS2D, dim ssm-ratio ×768
Linear ssm-ratio ×768 → 768
FFN mlp-ratio ×768 ] ×2 | VSSBlock(ssm-ratio=2, mlp-ratio=4)
[ Linear 768 → ssm-ratio ×768
DWConv 3×3, ssm-ratio ×768
SS2D, dim ssm-ratio ×768
Linear ssm-ratio ×768 → 768
FFN mlp-ratio ×768 ] ×2 | VSSBLock(ssm-ratio=2, mlp-ratio=4)
[ Linear 1024 → ssm-ratio ×1024
DWConv 3×3, ssm-ratio ×1024
SS2D, dim ssm-ratio ×1024
Linear ssm-ratio ×1024 → 1024
FFN mlp-ratio ×1024 ] ×2 |
| | 1×1 | average pool, 1000-d fc, softmax | | |
| Param. (M) | | 30.2 | 50.1 | 88.6 |
| FLOPs | | $4.91×10^9$ | $8.72×10^9$ | $15.36×10^9$ |

Table 5: Details of accelerating VMamba.

| Model | d_state | ssm-ratio | DWConv | multiculative branch | layers numbers | FFN | Params (M) | FLOPs (G) | TP. (img/s) | Train TP. (img/s) | Top-1 (%) |
|---|---|---|---|---|---|---|---|---|---|---|---|
| Vanilla-VMamba-T | 16 | 2.0 | ✓ | ✓ | [2,2,9,2] | | 22.9M | 5.63G | 426 | 138 | 82.17 |
| Step(a) | 16 | 2.0 | ✓ | ✓ | [2,2,9,2] | | 22.9M | 5.63G | 467 | 165 | 82.17 |
| Step(b) | 16 | 2.0 | ✓ | ✓ | [2,2,9,2] | | 22.9M | 5.63G | 464 | 184 | 82.17 |
| Step(c) | 16 | 2.0 | ✓ | ✓ | [2,2,9,2] | | 22.9M | 5.63G | 638 | 195 | 82.17 |
| Step(d) | 16 | 2.0 | | ✓ | [2,2,2,2] | ✓ | 29.0M | 5.63G | 813 | 248 | 81.65 |
| Step(d.1) | 16 | 1.0 | | ✓ | [2,2,2,2] | ✓ | 22.9M | 4.02G | 1336 | 405 | 81.05 |
| Step(d.2) | 16 | 1.0 | | ✓ | [2,2,5,2] | ✓ | 28.2M | 5.18G | 1137 | 348 | 82.24 |
| Step(e) | 16 | 1.0 | | | [2,2,5,2] | ✓ | 26.2M | 4.86G | 1179 | 360 | 82.17 |
| Step(e.1) | 16 | 1.0 | ✓ | | [2,2,5,2] | ✓ | 26.3M | 4.87G | 1164 | 358 | 82.31 |
| Step(e.2) | 1 | 1.0 | ✓ | | [2,2,5,2] | ✓ | 25.6M | 3.98G | 1942 | 647 | 81.87 |
| Step(f) | 1 | 2.0 | ✓ | | [2,2,5,2] | ✓ | 30.7M | 4.86G | 1340 | 464 | 82.49 |
| Step(g) | 1 | 1.0 | ✓ | | [2,2,8,2] | ✓ | 30.2M | 4.91G | 1686 | 571 | 82.60 |

the top-1 classification accuracy on ImageNet-1K. We also conduct experiments under the 'linear tuning' setting, where only the header network, consisting of a single linear module, is fine-tuned from random initialization using features extracted by the backbone models.

According to the results summarized in Table 9, VMamba demonstrates the most stable performance across (*i.e.*, modest performance drop) different input image sizes, achieving a top-1 classification accuracy of 74.7% without fine-tuning (79.2% with linear tuning), while maintaining a relatively high throughput of 149 images per second at an input resolution of 768 × 768. In comparison, Swin [36] achieves the second-highest performance with a top-1 accuracy of 73.1% without fine-tuning (77.5% under linear tuning) at the same input size, using scaled window sizes (set as the resolution divided by 32). However, its throughput significantly drops to 53 images per second. Furthermore, ConvNeXt [37] maintains a relatively high inference speed (*i.e.*, a throughput of 103 images per second) at the largest input resolution. However, its classification accuracy drops to 69.5% when directly tested on images of size 768 × 768, indicating its limited adaptability to images with large spatial resolutions. Deit-S also shows a dramatic performance drop, primarily due to the interpolation used in the absolute positional embedding.

Table 6: Performance comparison on ImageNet-1K with an image size of 224. [†] indicates that Vim is trained solely in float32 in practice, with a training throughput of 232. [69].

| Model | Image Size | Params (M) | FLOPs (G) | TP. (img/s) | Train TP. (img/s) | Top-1 (%) |
|---|---|---|---|---|---|---|
| DeiT-S [57] | $224^2$ | 22M | 4.6G | 1761 | 2404 | 79.8 |
| DeiT-B [57] | $224^2$ | 86M | 17.5G | 503 | 1032 | 81.8 |
| ConvNeXt-T [37] | $224^2$ | 29M | 4.5G | 1198 | 702 | 82.1 |
| ConvNeXt-S [37] | $224^2$ | 50M | 8.7G | 684 | 445 | 83.1 |
| ConvNeXt-B [37] | $224^2$ | 89M | 15.4G | 436 | 334 | 83.8 |
| HiViT-T [66] | $224^2$ | 19M | 4.6G | 1393 | 1304 | 82.1 |
| HiViT-S [66] | $224^2$ | 38M | 9.1G | 712 | 698 | 83.5 |
| HiViT-B [66] | $224^2$ | 66M | 15.8G | 456 | 544 | 83.8 |
| Swin-T [36] | $224^2$ | 28M | 4.5G | 1244 | 987 | 81.3 |
| Swin-S [36] | $224^2$ | 50M | 8.7G | 718 | 642 | 83.0 |
| Swin-B [36] | $224^2$ | 88M | 15.5G | 458 | 496 | 83.5 |
| XCiT-S12/16 | $224^2$ | 26M | 4.9G | 1283 | 935 | 82.0 |
| XCiT-S24/16 | $224^2$ | 48M | 9.2G | 671 | 509 | 82.6 |
| XCiT-M24/16 | $224^2$ | 84M | 16.2G | 423 | 385 | 82.7 |
| S4ND-ConvNeXt-T [44] | $224^2$ | 30M | 5.2G | 683 | 369 | 82.2 |
| S4ND-ViT-B [44] | $224^2$ | 89M | 17.1G | 398 | 400 | 80.4 |
| Vim-S [69] | $224^2$ | 26M | 5.3G | 811 | 344[†] | 80.5 |
| Vanilla-VMamba-T | $224^2$ | 23M | 5.6G | 638 | 195 | 82.2 |
| Vanilla-VMamba-S | $224^2$ | 44M | 11.2G | 359 | 111 | 83.5 |
| Vanilla-VMamba-B | $224^2$ | 76M | 18.0G | 268 | 84 | 83.7 |
| VMamba-T | $224^2$ | 30M | 4.9G | 1686 | 571 | 82.6 |
| VMamba-S[$s2l15$] | $224^2$ | 50M | 8.7G | 877 | 314 | 83.6 |
| VMamba-B[$s2l15$] | $224^2$ | 89M | 15.4G | 646 | 247 | 83.9 |
| VMamba-S[$s1l20$] | $224^2$ | 49M | 8.6G | 1106 | 390 | 83.3 |
| VMamba-B[$s1l20$] | $224^2$ | 87M | 15.2G | 827 | 313 | 83.8 |

Notably, VMamba displays a linear increase in computational complexity, as measured by FLOPs, which is comparable to CNN-based architectures. This finding aligns with the theoretical conclusions drawn from selective SSMs [17].

# H Ablation Study

## H.1 Influence of the Scanning Pattern

In the main submission, we validate the effectiveness of the proposed scanning pattern (referred to as Cross-Scan) in SS2D by comparing it to three alternative image traversal approaches, *i.e.*, Unidi-Scan, Bidi-Scan, and Cascade-Scan (Figure 12). Notably, since Unidi-Scan, Bidi-Scan, and Cross-Scan are all implemented in `Triton`, they exhibit minimal differences in throughput. The results in Table 10 indicate that Cross-Scan demonstrates superior data modeling capacity, as reflected by its higher classification accuracy. This advantage likely stems from the two-dimensional prior introduced by the four-way scanning design. Nevertheless, the practical implementation of Cascade-Scan is significantly constrained by its relatively slow computational pace, primarily due to the inadequate compatibility between selective scanning and high-dimensional data, which is further affected by the multi-step scanning procedure.

Figure 13 indirectly demonstrates that among the analyzed scanning methods, only Bidi-Scan, Cascade-Scan, and Cross-Scan showcase global ERFs. Moreover, only Cross-Scan and Cascade-Scan exhibit two-dimensional (2D) priors. It is also worth noting that `DWConv` [23] plays a critical role in establishing 2D priors, thereby contributing to the formation of global ERFs.

## H.2 Influence of the Initialization Approach

In our study, we adopted the initialization scheme originally proposed for the SS2D block in S4D [19]. Therefore, it is necessary to investigate the contribution of this initialization method to the effective-

Table 7: Object detection and instance segmentation results on COCO dataset. FLOPs are calculated using inputs of size $1280 \times 800$. Here, $AP^b$ and $AP^m$ denote box AP and mask AP, respectively. "1×" indicates models fine-tuned for 12 epochs, while "3×MS" signifies the utilization of multi-scale training for 36 epochs.

| Backbone | $AP^b$ | $AP^b_{50}$ | $AP^b_{75}$ | $AP^m$ | $AP^m_{50}$ | $AP^m_{75}$ | Params | FLOPs |
|---|---|---|---|---|---|---|---|---|
| **Mask R-CNN 1× schedule** | | | | | | | | |
| Swin-T | 42.7 | 65.2 | 46.8 | 39.3 | 62.2 | 42.2 | 48M | 267G |
| ConvNeXt-T | 44.2 | 66.6 | 48.3 | 40.1 | 63.3 | 42.8 | 48M | 262G |
| Vanilla-VMamba-T | 46.5 | 68.5 | 50.7 | 42.1 | 65.5 | 45.3 | 42M | 286G |
| VMamba-T | 47.3 | 69.3 | 52.0 | 42.7 | 66.4 | 45.9 | 50M | 271G |
| Swin-S | 44.8 | 66.6 | 48.9 | 40.9 | 63.4 | 44.2 | 69M | 354G |
| ConvNeXt-S | 45.4 | 67.9 | 50.0 | 41.8 | 65.2 | 45.1 | 70M | 348G |
| Vanilla-VMamba-S | 48.2 | 69.7 | 52.5 | 43.0 | 66.6 | 46.4 | 64M | 400G |
| VMamba-S | 48.7 | 70.0 | 53.4 | 43.7 | 67.3 | 47.0 | 70M | 349G |
| Swin-B | 46.9 | – | – | 42.3 | – | – | 107M | 496G |
| ConvNeXt-B | 47.0 | 69.4 | 51.7 | 42.7 | 66.3 | 46.0 | 108M | 486G |
| Vanilla-VMamba-B | 48.6 | 70.0 | 53.1 | 43.3 | 67.1 | 46.7 | 96M | 540G |
| VMamba-B | 49.2 | 71.4 | 54.0 | 44.1 | 68.3 | 47.7 | 108M | 485G |
| **Mask R-CNN 3× MS schedule** | | | | | | | | |
| Swin-T | 46.0 | 68.1 | 50.3 | 41.6 | 65.1 | 44.9 | 48M | 267G |
| ConvNeXt-T | 46.2 | 67.9 | 50.8 | 41.7 | 65.0 | 44.9 | 48M | 262G |
| Vanilla-VMamba-T | 48.5 | 70.0 | 52.7 | 43.2 | 66.9 | 46.4 | 42M | 286G |
| VMamba-T | 48.8 | 70.4 | 53.5 | 43.7 | 67.4 | 47.0 | 50M | 271G |
| Swin-S | 48.2 | 69.8 | 52.8 | 43.2 | 67.0 | 46.1 | 69M | 354G |
| ConvNeXt-S | 47.9 | 70.0 | 52.7 | 42.9 | 66.9 | 46.2 | 70M | 348G |
| Vanilla-VMamba-S | 49.7 | 70.4 | 54.2 | 44.0 | 67.6 | 47.3 | 64M | 400G |
| VMamba-S | 49.9 | 70.9 | 54.7 | 44.2 | 68.2 | 47.7 | 70M | 349G |

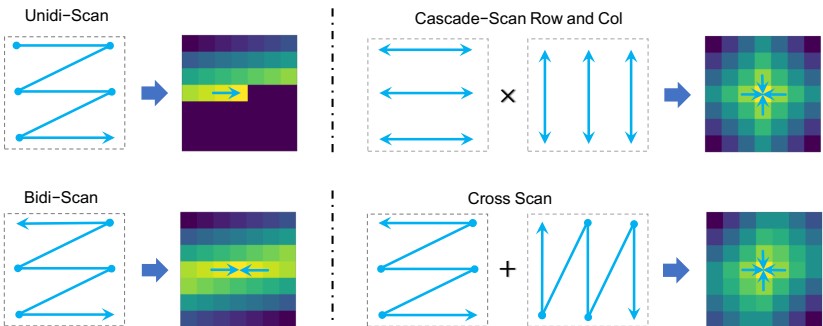

Figure 12: Illustration of different scanning patterns for selective scan.

ness of VMamba. To explore this further, we replaced the default initialization with two alternative methods: random initialization and zero initialization.

For both initialization methods, we set the parameter $\mathbf{D}$ in equation 1 to a vector of all ones, mimicking a basic skip connection (thus we have $\mathbf{y} = \mathbf{Ch} + \mathbf{Du}$). Additionally, the weights and biases associated with the transformation to the dimension $D_v$ (which matches the input size), are initialized as random vectors. In contrast, Mamba [17] employs a more sophisticated initialization.

Table 8: Semantic segmentation results on ADE20K using UperNet [63]. We evaluate the performance of semantic segmentation on the ADE20K dataset with UperNet [63]. FLOPs are calculated with input sizes of $512 \times 2048$. "SS" and "MS" denote single-scale and multi-scale testing, respectively.

| method | crop size | mIoU (SS) | mIoU (MS) | Params | FLOPs |
|---|---|---|---|---|---|
| Swin-T | $512^2$ | 44.5 | 45.8 | 60M | 945G |
| ConvNeXt-T | $512^2$ | 46.0 | 46.7 | 60M | 939G |
| Vanilla-VMamba-T | $512^2$ | 47.3 | 48.3 | 55M | 964G |
| VMamba-T | $512^2$ | 48.0 | 48.8 | 62M | 949G |
| Swin-S | $512^2$ | 47.6 | 49.5 | 81M | 1039G |
| ConvNeXt-S | $512^2$ | 48.7 | 49.6 | 82M | 1027G |
| Vanilla-VMamba-S | $512^2$ | 49.5 | 50.5 | 76M | 1081G |
| VMamba-S | $512^2$ | 50.6 | 51.2 | 82M | 1028G |
| Swin-B | $512^2$ | 48.1 | 49.7 | 121M | 1188G |
| ConvNeXt-B | $512^2$ | 49.1 | 49.9 | 122M | 1170G |
| Vanilla-VMamba-B | $512^2$ | 50.0 | 51.3 | 110M | 1226G |
| VMamba-B | $512^2$ | 51.0 | 51.6 | 122M | 1170G |

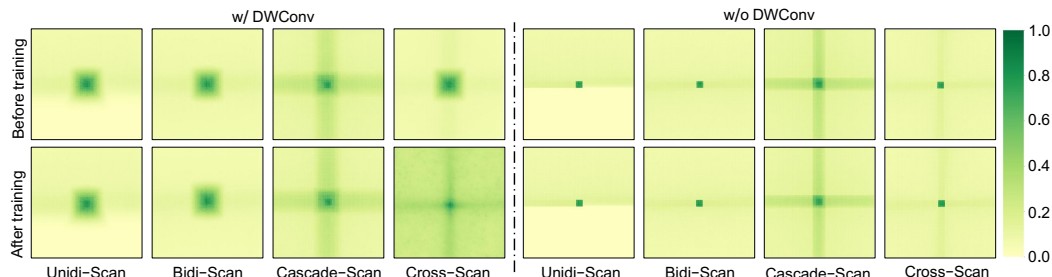

Figure 13: The visualization of ERF for models with different scanning patterns.

The main distinction between random and zero initialization lies in the parameter $A$ in equation 6, which is typically initialized as a HiPPO matrix in both Mamba [17, 19] and our implementation of VMamba. Given that we selected the hyper-parameter `d_state` to be 1, the Mamba initialization for $log(A)$ can be simplified to all zeros, which aligns with zero initialization. In contrast, random initialization assigns a random vector to $log(A)$. We choose to initialize $log(A)$ rather than $A$ directly to keep $A$ near the all-ones matrix when the network parameters are close to zero, which empirically enhances the training stability.

The experimental results in Table 11 indicate that, at least for image classification with SS2D blocks, the model's performance is not significantly affected by the initialization method. Therefore, within this context, the sophisticated initialization method employed in Mamba [17] can be substituted with a simpler, more straightforward approach. We also visualize the ERF maps of models trained with different initialization methods (see Figure 14), which intuitively reflect SS2D's robustness across various initialization schemes.

## H.3 Influence of the `d_state` Parameter

Throughout this study, we primarily set the value of `d_state` to 1 to optimize VMamba's computational speed. To further explore the impact of `d_state` on the model's performance, we conduct a series of experiments.

As shown in Table 12, with all other hyper-parameters fixed, we increase `d_state` from 1 to 4. This results in a slight improvement in performance but a substantial decrease in throughput, indicating a significant negative impact on the VMamba's computational efficiency. However, increasing `d_state`

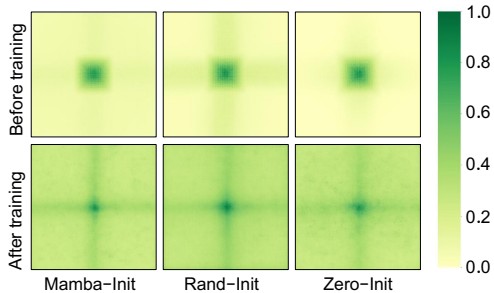

Figure 14: The visualization of ERF of VMamba with different initialization.

to 8, while reducing `ssm-ratio` to maintain computational complexity, leads to improved accuracy. Moreover, when `d_state` is further increased to 16, with `ssm-ratio` set to 1, performance declines.

These findings suggest that modest increases in `d_state` may not necessarily lead to better performance. Instead, selecting the optimal combination of `d_state` and `ssm-ratio` is crucial for achieving a good trade-off between inference speed and performance.

### H.4 Influence of `ssm-ratio`, `mlp-ratio`, and `layer numbers`

In this section, we investigate the trade-offs among `ssm-ratio`, `layer numbers`, and `mlp-ratio`.

Experimental results shown in Table 13 indicate that reducing `ssm-ratio` significantly decreases performance but substantially improves inference speed. Conversely, increasing `layer numbers` enhances the performance while slowing down the model.

As the hyper-parameter `ssm-ratio` represents the dimension used by the SS2D module, the trade-off between `ssm-ratio` and `layer numbers` can be interpreted as a balance between `channel-mixing` and `token-mixing` [56]. Furthermore, we reduce `mlp-ratio` from 4.0 to 2.0 and progressively increase `ssm-ratio` to maintain constant FLOPs, as shown in Table 14. The results presented in Tables 13 and 14 highlight the importance of an optimal combination of `ssm-ratio`, `mlp-ratio`, and `layer numbers` for constructing a model that balances effectiveness and efficiency.

### H.5 Influence of the Activation Function

In VMamba, the SiLU [14] activation function is utilized to build the SS2D block. However, experimental results in Table 15 show that VMamba maintains robustness across different activation functions. This implies that the choice of activation function does not substantially affect the model's performance. Therefore, there is flexibility to choose an appropriate activation function based on computational constraints or other preferences.

Table 9: Comparison of generalizability to inputs with increased spatial resolutions. The throughput and training throughput are measured with a batch size of 32 using PyTorch 2.0 on an A100 GPU paired with an AMD EPYC 7542 CPU. Unlike throughput, the model's forward pass, loss calculation, and backward pass are included in calculating the training throughput, with mixed precision. We re-implemented the HiViT-T, as the checkpoint for HiViT-T has not been released. † denotes that the batch size $\leq 16$ due to out-of-memory (OOM) issues.

| Model | Image Size | Param. (M) | FLOPs (G) | TP. (img/s) | Train TP. (img/s) | Top-1 acc. (%) | Top-1 acc. (%) (w/ linear tuning) |
|---|---|---|---|---|---|---|---|
| **SSM-Based** | | | | | | | |
| VMamba-Tiny | $224^2$ | 30M | 4.91G | 1490 | 418 | 82.60 | 82.64 |
| | $288^2$ | 30M | 8.11G | 947 | 303 | 82.95 | 83.03 |
| | $384^2$ | 30M | 14.41G | 566 | 187 | 82.41 | 82.77 |
| | $512^2$ | 30M | 25.63G | 340 | 121 | 80.92 | 81.88 |
| | $640^2$ | 30M | 40.04G | 214 | 75 | 78.60 | 80.62 |
| | $768^2$ | 30M | 57.66G | 149 | 53 | 74.66 | 79.22 |
| VMamba-Tiny[$s2l5$] | $224^2$ | 31M | 4.86G | 1227 | 399 | 82.49 | 82.52 |
| | $288^2$ | 31M | 8.03G | 761 | 255 | 82.81 | 82.93 |
| | $384^2$ | 31M | 14.27G | 452 | 155 | 82.51 | 82.74 |
| | $512^2$ | 31M | 25.38G | 272 | 100 | 81.07 | 82.02 |
| | $640^2$ | 31M | 39.65G | 170 | 60 | 79.30 | 81.02 |
| | $768^2$ | 31M | 57.09G | 117 | 42 | 76.06 | 79.69 |
| Vanilla-VMamba-Tiny | $224^2$ | 23M | 5.63G | 628 | 189 | 82.17 | 82.09 |
| | $288^2$ | 23M | 9.30G | 390 | 117 | 82.74 | 82.76 |
| | $384^2$ | 23M | 16.53G | 212 | 65 | 82.40 | 82.72 |
| | $512^2$ | 23M | 29.39G | 138 | 53 | 81.05 | 81.97 |
| | $640^2$ | 23M | 45.93G | 87 | 27 | 78.79 | 80.71 |
| | $768^2$ | 23M | 66.14G | 52 | 18 | 75.09 | 79.12 |
| **Transformer-Based** | | | | | | | |
| Swin-Tiny | $224^2$ | 28M | 4.51G | 1142 | 769 | 81.19 | 81.18 |
| | $288^2$ | 28M | 7.60G | 638 | 489 | 81.46 | 81.62 |
| | $384^2$ | 28M | 14.05G | 316 | 268 | 80.67 | 81.12 |
| | $512^2$ | 28M | 26.65G | 176 | 131 | 78.97 | 80.21 |
| | $640^2$ | 28M | 45.00G | 88 | 68 | 76.55 | 78.89 |
| | $768^2$ | 29M | 70.72G | 53 | 38 | 73.06 | 77.54 |
| XCiT-S12/16 | $224^2$ | 26M | 4.87G | 1127 | 505 | 81.87 | 81.89 |
| | $288^2$ | 26M | 8.05G | 724 | 462 | 82.44 | 82.44 |
| | $384^2$ | 26M | 14.31G | 425 | 308 | 81.84 | 82.21 |
| | $512^2$ | 26M | 25.44G | 244 | 185 | 79.80 | 80.92 |
| | $640^2$ | 26M | 39.75G | 158 | 122 | 76.84 | 79.00 |
| | $768^2$ | 26M | 57.24G | 111 | 87 | 72.52 | 76.92 |
| HiViT-Tiny | $224^2$ | 19M | 4.60G | 1261 | 1041 | 81.92 | 81.85 |
| | $288^2$ | 19M | 7.93G | 750 | 614 | 82.45 | 82.42 |
| | $384^2$ | 19M | 15.21G | 388 | 333 | 81.51 | 81.91 |
| | $512^2$ | 20M | 30.56G | 186 | 150 | 79.30 | 80.49 |
| | $640^2$ | 20M | 54.83G | 93 | 71† | 76.09 | 78.58 |
| | $768^2$ | 20M | 91.41G | 55 | 37† | 71.38 | 76.47 |
| DeiT-Small | $224^2$ | 22M | 4.61G | 1573 | 1306 | 80.69 | 80.40 |
| | $288^2$ | 22M | 7.99G | 914 | 1124 | 80.80 | 80.63 |
| | $384^2$ | 22M | 15.52G | 502 | 697 | 78.87 | 79.54 |
| | $512^2$ | 22M | 31.80G | 261 | 387 | 74.21 | 76.91 |
| | $640^2$ | 23M | 58.17G | 149 | 244 | 68.04 | 73.31 |
| | $768^2$ | 23M | 98.70G | 90 | 156 | 60.98 | 69.62 |
| **ConvNet-Based** | | | | | | | |
| ConvNeXt-Tiny | $224^2$ | 29M | 4.47G | 1107 | 614 | 82.05 | 81.95 |
| | $288^2$ | 29M | 7.38G | 696 | 403 | 82.23 | 82.30 |
| | $384^2$ | 29M | 13.12G | 402 | 240 | 81.05 | 81.78 |
| | $512^2$ | 29M | 23.33G | 226 | 140 | 78.03 | 80.37 |
| | $640^2$ | 29M | 36.45G | 147 | 90 | 74.27 | 78.77 |
| | $768^2$ | 29M | 52.49G | 103 | 63 | 69.50 | 76.89 |

Table 10: The performance of VMamba-T with different scanning patterns.

| Model | Params (M) | FLOPs (G) | TP. (img/s) | Train TP. (img/s) | Top-1 (%) |
|---|---|---|---|---|---|
| **VMamba w/ dwconv** | | | | | |
| Unidi-Scan | 30.2M | 4.91G | 1682 | 571 | 82.26 |
| Bidi-Scan | 30.2M | 4.91G | 1687 | 572 | 82.49 |
| Cascade-Scan | 30.2M | 4.91G | 998 | 308 | 82.42 |
| Cross-Scan | 30.2M | 4.91G | 1686 | 571 | 82.60 |
| **VMamba w/o dwconv** | | | | | |
| Unidi-Scan | 30.2M | 4.89G | 1716 | 578 | 80.88 |
| Bidi-Scan | 30.2M | 4.89G | 1719 | 578 | 81.80 |
| Cascade-Scan | 30.2M | 4.90G | 1007 | 309 | 81.71 |
| Cross-Scan | 30.2M | 4.89G | 1717 | 577 | 82.25 |

Table 11: The performance of VMamba-T with different initialization.

| initialization | Params (M) | FLOPs (G) | TP. (img/s) | Train TP. (img/s) | Top-1 acc. (%) |
|---|---|---|---|---|---|
| mamba | 30.2 | 4.91 | 1686 | 571 | 82.60 |
| rand | 30.2 | 4.91 | 1682 | 570 | 82.58 |
| zero | 30.2 | 4.91 | 1683 | 570 | 82.67 |

Table 12: The performance of VMamba-T with different d_state.

| d_state | ssm-ratio | Params (M) | FLOPs (G) | TP. (img/s) | Train TP. (img/s) | Top-1 acc. (%) |
|---|---|---|---|---|---|---|
| 1 | 2.0 | 30.7 | 4.86 | 1340 | 464 | 82.49 |
| 2 | 2.0 | 30.8 | 4.98 | 1269 | 432 | 82.50 |
| 4 | 2.0 | 31.0 | 5.22 | 1147 | 382 | 82.51 |
| 8 | 1.5 | 28.6 | 5.04 | 1148 | 365 | 82.69 |
| 16 | 1.0 | 26.3 | 4.87 | 1164 | 358 | 82.31 |

Table 13: The performance of VMamba-T under different combination of ssm-ratio and layer numbers.

| ssm-ratio | layer numbers | Params (M) | FLOPs (G) | TP. (img/s) | Train TP. (img/s) | Top-1 acc. (%) |
|---|---|---|---|---|---|---|
| 2.0 | [2,2,5,2] | 30.7 | 4.86 | 1340 | 464 | 82.49 |
| 1.0 | [2,2,5,2] | 25.6 | 3.98 | 1942 | 647 | 81.87 |
| 1.0 | [2,2,8,2] | 30.2 | 4.91 | 1686 | 571 | 82.60 |

Table 14: The performance of VMamba under different combination of ssm-ratio and mlp-ratio.

| mlp-ratio | ssm-ratio | Params (M) | FLOPs (G) | TP. (img/s) | Train TP. (img/s) | Top-1 acc. (%) |
|---|---|---|---|---|---|---|
| 4.0 | 1.0 | 30.2 | 4.91 | 1686 | 571 | 82.60 |
| 3.0 | 1.5 | 28.5 | 4.65 | 1419 | 473 | 82.75 |
| 2.0 | 2.5 | 29.9 | 4.95 | 1075 | 361 | 82.86 |

Table 15: The performance of VMamba-T with different activation functions in SS2D.

| activation | Params (M) | FLOPs (G) | TP. (img/s) | Train TP. (img/s) | Top-1 acc. (%) |
|---|---|---|---|---|---|
| SiLU | 30.2 | 4.91 | 1686 | 571 | 82.60 |
| GELU | 30.2 | 4.91 | 1680 | 570 | 82.53 |
| ReLU | 30.2 | 4.91 | 1684 | 577 | 82.65 |

