# OpenReview forum: "VMamba: Visual State Space Model"
_NeurIPS.cc/2024/Conference — NeurIPS 2024 spotlight_

### Official Review · Reviewer_c4kg · 2024-07-02

**Soundness:** 4
**Presentation:** 4
**Contribution:** 3
**Rating:** 7
**Confidence:** 5

**Summary:**

This paper presents VMamba, a novel vision backbone model inspired by the famous Mamba state-space sequence model. The main contribution of VMamba is its ability to achieve efficient visual representation learning with linear computational complexity. The core of VMamba is the VSS block, which incorporates the 2D-Selective-Scan module (SS2D), thereby extending the Mamba model that is a 1D selective scan good for NLP tasks. With SS2D, we can work nicely with inductive biases associated with 2D image space.

VMamba's architecture consists of multiple stages with hierarchical representations (similar to ViT). The authors introduce three model sizes: Tiny, Small, and Base. The VSS blocks replace the S6 module from Mamba with the SS2D module, and further enhancements are made by eliminating unnecessary components and optimizing the architecture for better computational efficiency - using the Triton language.

Extensive experiments demonstrate VMamba's promising performance across various visual perception tasks, including image classification on ImageNet-1K, object detection, instance segmentation on MSCOCO, and semantic segmentation on ADE20K. VMamba consistently achieves superior accuracy and throughput compared to existing benchmark models, showcasing its scalability and adaptability to different input resolutions and downstream tasks.

**Strengths:**

* 2D-Selective-Scan Module: The introduction of the 2D-Selective-Scan (SS2D) module is a creative solution to bridge the gap between 1D selective scan and 2D vision data.
* Comprehensive Experiments: The paper provides extensive experimental results on multiple benchmarks, including ImageNet-1K, MSCOCO, and ADE20K, demonstrating the effectiveness and robustness of VMamba across various tasks.
* Clear Explanation: The paper is well-written, with clear explanations. The authors provide detailed descriptions of the architecture, modules, and experimental setups, making it accessible to readers.
* Visualization: The use of visualizations, such as activation maps and effective receptive fields (ERF), helps in understanding the SS2D mechanism and the model's behavior, which is very important part of all the ablation studies.
* Impact on Visual Representation Learning: VMamba addresses a critical issue in vision models by reducing computational complexity from quadratic to linear, which can significantly impact the field of visual representation learning.

**Weaknesses:**

* Limited Comparison with Other SSM-based Models: While the paper does compare VMamba with several benchmark models, it would benefit from a more detailed comparison with other state-space models (SSM) in the vision domain. Specifically, models like S4ND and Vim are mentioned, but the comparisons are somewhat brief. Providing more in-depth analysis and results would strengthen the argument for VMamba's superiority.
* Adding more interesting works to Related Work section: There are some interesting works on neuromorphic vision and processing with SSMs that authors should cite and mention:

[1] State Space Models for Event Cameras. Nikola Zubić, Mathias Gehrig, Davide Scaramuzza - CVPR 2024, Spotlight

[2] Scalable Event-by-event Processing of Neuromorphic Sensory Signals With Deep State-Space Models. Mark Schöne, Neeraj Mohan Sushma, Jingyue Zhuge, Christian Mayr, Anand Subramoney, David Kappel - ICONS 2024

* Generalization to Other Tasks: The experiments focus mainly on standard benchmarks for image classification, object detection, and segmentation. However, it is not clear how well VMamba generalizes to other types of visual tasks such as video analysis, 3D vision, or more complex scene understanding. Including some preliminary results or at least discussions on these aspects could highlight the versatility of VMamba further.
* Clarity in Mathematical Derivations: Some of the mathematical derivations, especially in the relationship between SS2D and self-attention, are complex and may not be easily accessible to all readers. Simplifying the explanations or providing more intuitive visual insights alongside the formal derivations could enhance understanding. Also, they are not rigorously mathematically proven.

**Questions:**

- Could you provide more detailed comparisons with other state-space models (SSMs) used in the vision domain, such as S4ND and Vim? Specifically, how does VMamba perform in terms of accuracy, computational efficiency, and memory usage compared to these models?
- How well does VMamba generalize to other types of visual tasks beyond image classification, object detection, and segmentation? Have you considered evaluating VMamba on tasks such as video analysis, 3D vision, or more complex scene understanding? How does it scale on these tasks?
- How sensitive is VMamba to various hyperparameters? It would be helpful to know if specific hyperparameters are critical to achieving the reported performance and if there are guidelines or best practices for tuning them.
- How easily can VMamba be integrated into existing deep learning frameworks and pipelines? Are there any specific requirements or modifications needed for seamless integration?

**Limitations:**

Authors addressed everything regarding limitations section.

---

> ### Author Rebuttal · Authors · 2024-08-06
>
> # Response to Reviewer c4kg
>
> We appreciate the reviewer’s thoughtful review and constructive comments. In our responses, we address the following concerns: a detailed comparison with SSM-based methods, the generalizability of VMamba, sensitivity to hyper-parameters, and potential for integration into various frameworks.
>
> ### **Detailed Comparison with SSM-based Methods**
>
> In Table 1 of the main submission, we have already compared our method to S4ND [2] and Vim [4] in terms of the number of parameters, train throughput, and the Top-1 accuracy on ImageNet-1K. To provide a more comprehensive evaluation, we additionally conduct comparison on FLOPs and the memory usage, and the results are reported in the following table.
>
> Moreover, we also compare the performance (both effectiveness and efficiency) change with increasing input resolution in Figure 1 in the `attachment`. For qualitative comparison, we visualize the ERF of S4ND and Vim, and the results are shown in Figure 2 in the `attachment`.  We will include these results and the associated analysis in the revised manuscript.
>
> | Model | Hierarchical | Params (M) | FLOPs (G) | TP. (img/s) | Test Mem. (M) | Train TP. (img/s) | Train Mem. (M) | Top-1 (\%) |
> |:---:|:---:|:---:|:---:|:---:|:---:|:---:|:---:|:---:|
> |DeiT-S      |False |22M |4.6G  |1761 |582  |2404 |4562  |79.8 |
> |DeiT-B      |False |86M |17.5G |503  |1032 |1404 |9511  |81.8 |
> |S4ND-ViT-B  |False |89M |17.1G |398  |2221 |400  |15868 |80.4 |
> |Vim-S       |False |26M |5.3G  |811  |1055 |344 $\dagger$ (232) |9056 $\dagger$ (16150) |80.5 |
> |Swin-T      |True  |28M |4.5G  |1244 |3092 |987  |9798  |81.3 |
> |ConvNeXt-T  |True  |29M |4.5G  |1198 |2498 |702  |9450  |82.1 |
> |S4ND-Conv-T |True  |30M |5.2G  |683  |3945 |369  |18843 |82.2 |
> |Vanilla-VMamba-T |True  |23M |5.6G  |638  |6042 |195  |16452 |82.2 |
> |VMamba-T    |True  |30M |4.9G  |1686 |3064 |571  |12394 |82.6 |
>
> [Performance comparison between VMamba and benchmark methods. $\dagger$ indicates the value is measured with mix-resolution while Vim does not support training with mix-resolution (values in the brackets are results obtained with fp32).]
>
> ### **Additional Related Studies**
> We thank the reviewer for bringing these inspiring studies to our attention. We will include references to these papers in the revised version.
>
> ### **Versatility of VMamba**
> Due to our limited computational resources, we have focused on conducting experiments on benchmark tasks in vision modeling. However, we recognize the importance of illustrating the potential of the proposed method in more generalized tasks.
>
> A preliminary literature review of recently proposed SSM-based approaches in vision tasks, along with our private communications with researchers in the field, highlights the potential of the 2D selective scan technique (SS2D) introduced in this study. SS2D does not make specific assumptions about the layout or modality of the input data, which allows it to be generalized to various tasks. For example, SS2D can process video data by traversing a spatial-temporal plane of frame patches. To our knowledge, recent studies leveraging scanning patterns analogous to SS2D have shown success in various tasks, including image restoration and multimodal data understanding, in addition to those mentioned in the question. We will add these results to the final version and cite their works if they are published by then.
>
> Despite not being inherently prohibited, we anticipate challenges in directly migrating SS2D to diverse downstream tasks due to varying requirements. Bridging the gap between SS2D and these tasks, along with proposing a more generalized scanning pattern for vision tasks, is a promising research direction. We will include this discussion in the revised version, hopefully to provide readers with some inspiration.
>
> ### **Clarity in Mathematical Derivations**
>
> Due to limited space, we have included detailed proofs in the appendix and will provide more rigorous and clearer derivations in the revised version. We also recognize the significance of providing more intuitive and accessible explanations, and will include them in the revised version.
>
> ### **Sensitivity ot Hyper-parameters**
> According to our experience, we have not found any hyperparameter to which VMamba is particularly sensitive. This observation is also supported by the ablation results on single hyper-parameters (initialization approach in Table 11 and activation function in Table 15) as well as different combinations (Tables 12, 13, and 14) included in the Appendix.
>
> We conducted additional experiments on the influence of the learning rate, and the results are reported in the following table. We will include this discussion in the revised version.
>
> |Model       |Params (M) |FLOPs (G) |lr | Top 1. (\%) |
> |:--:|:--:|:--:|:--:|:--:|
> |VMamba-Tiny        |30M |4.91G |2e-3 |82.70 |
> |VMamba-Tiny $\dagger$ |30M |4.91G |1e-3 |82.62 |
> |VMamba-Tiny        |30M |4.91G |5e-4 |82.16 |
>
> [The performance of VMamba-T with different learning rate. Results marked by $\dagger$ is the default setting used in the submission. All the models here are trained on `[SERVER 2]`.]
>
> ### **Potential of Integrating into Various Frameworks**
>
> The core of VMamba lies in the design of the SS2D module, which aims to bridge the gap between 1D sequence scanning and 2D plane traversing, rather than specific architectural configurations. SS2D can function as an end-to-end token mixer, allowing it to be integrated into various mainstream backbone networks in computer vision.
>
> Indeed, integrating SS2D into existing frameworks requires additional considerations. One critical aspect is the numerical precision settings in the model, which significantly impact performance and computational speed. Another important factor is the inclusion of normalization layers to stabilize the training process. We will include these points in the revised version to assist researchers who may want to build upon our work.

---

> > ### Comment · Reviewer_c4kg · 2024-08-08
> >
> > 1. Authors did Detailed Comparison with SSM-based Methods along with experiments.
> > 2. They will include works of Zubić et al. and Schoene et al. in the related works section.
> > 3. Authors said that they will discuss more the generalized scanning pattern for vision tasks in the paper as future work, which is very interesting.
> > 4. Authors "have not found any hyperparameter to which VMamba is particularly sensitive".
> > 5. Pretty robust model to the changes in hyperparameters, they did experiments, for example learning rates, which is great.
> >
> > Given that the authors have addressed all my concerns with clear and effective experimental evidence, I am updating my score from Accept (7) to Strong Accept (8).

---

### Official Review · Reviewer_ZfkW · 2024-07-12

**Soundness:** 4
**Presentation:** 4
**Contribution:** 3
**Rating:** 8
**Confidence:** 4

**Summary:**

This paper transplants the Mamba (Selective State Space Model), a linear complexity model originally designed for 1D language processing, into VMamba to process image data. It introduces the 2D selective scan and various acceleration techniques to facilitate the modeling of 2D data and enhance the speed of the network. The proposed VMamba model is trained and evaluated on a number of representative downstream tasks including ImageNet-1K classification, COCO object detection, and ASE20K semantic segmentation, and it is compared with strong baselines. A range of analyses and visualizations on the theoretical perspectives, design choices, and behavior of the model are also presented.

**Strengths:**

1.	VMamba is one of the first papers to attempt using Mamba, one of the most efficient and performant linear complexity models to date, to learn visual data and demonstrate effectiveness.
2.	The paper proposes a series of innovations to adapt the original Mamba's 1D sequential scanning to process 2D image data (SS2D) and increase the model's processing speed (image throughput) without compromising performance.
3.	In-depth deductions, comprehensive analyses, experiments, and visualizations on design choices, theoretical aspects (e.g., the relationship between SSM and Self-Attention), and model behaviors have been presented, carrying a huge volume of insightful findings that are valuable for future research.
4.	The proposed VMamba is compared to representative downstream tasks, including ImageNet-1K classification, COCO object detection, and ASE20K semantic segmentation. It shows comparable or better (and consistent) results to strong baselines (e.g., Swin, DeiT, and the concurrent Vim) and superior efficiency.
5.	As a general and simple visual model, the proposed VMamba potentially carries huge extension and generalization potential, which could inspire and impact a wide range of visual research.

**Weaknesses:**

I didn’t find any critical weakness in this paper. Apart from some limitations that have already been mentioned by the authors, such as large-scale experiments, training strategies, and hyperparameter search, the only part that I hope the paper can show more results is the ablation of some design choices. For instance, the performance change of removing the entire multiplicative branch, where Table 5 does not show a straight ablation because of more than 1 variable change. This problem also exists in some other tables of some other design choices and hyperparameters. But again, I think neither of these weaknesses is critical.

**Questions:**

Could the authors explain more on the statement in Lines 153-154, “such modification prevents the weights from being input-independent, resulting in a limited capacity for capturing contextual information”?

Others: Repetitive reference entries: [50] and [51], [50] and [60].

**Limitations:**

The authors adequately discuss the limitations and potential societal impact of this work. This paper also points out several potential improvements and future directions, with which I highly agree.

---

> ### Author Rebuttal · Authors · 2024-08-06
>
> # Response to Reviewer ZfkW
>
> We appreciate the reviewer’s thoughtful review and positive comments about our study. In the following sections, we address the reviewer’s primary concern regarding the lack of ablation on design choices and clarify several other issues raised.
>
> ### **More Ablation on Design Choices**
>
> First of all, we would like to clarify that our primary reason for modifying multiple hyper-parameters simultaneously is to ensure that the number of parameters and FLOPs remain comparable, facilitating a fair comparison between different model variants. In Table 5 (corresponding to Figure 3 (e) in Section 4.3 of the main submission), we detail the configurations used to optimize the overall performance of VMamba, balancing both effectiveness and efficiency rather than isolating the impact of each hyperparameter.
>
> However, we sincerely acknowledge the importance of analyzing the significance of each individual hyperparameter and architectural design choice on the overall performance. We plan to conduct more comprehensive experiments to extend the results of experiments isolating each hyperparameter in Table 5, and include those results in future versions of this study.
>
> To address the issue mentioned in the reviewer's comment, we have conducted additional experiments to analyze the influence of changing a single variable. The results are reported in the following table. Values for Step (e.1) and Step (e.2) are copied from Table 12 and Table 14 in the appendix, respectively, while Step (d.1) and Step (d.2) present new results obtained during the rebuttal process.
>
> | Model | d\_state | ssm\_ratio | DWConv | Multiplicative Branch | Layers | FFN |  Params (M) | FLOPs (G) | TP.  (img/s) | Train TP. (img/s)| Top-1 (\%) |
> |:--:|:--:|:--:|:--:|:--:|:--:|:--:|:--:|:--:|:--:|:--:|:--:|
> |Vanilla-VMamba-T|16 |2.0 |True |True |[2,2,9,2] |False|22.9M |5.63G |426        |138       |82.17        |
> |Step(a)         |16 |2.0 |True |True |[2,2,9,2] |False|22.9M |5.63G |467        |165       |82.17        |
> |Step(b)         |16 |2.0 |True |True |[2,2,9,2] |False|22.9M |5.63G |464        |184       |82.17        |
> |Step(c)         |16 |2.0 |True |True |[2,2,9,2] |False|22.9M |5.63G |638        |195       |82.17        |
> |Step(d)         |16 |2.0 |False|True |[2,2,2,2] |True |29.0M |5.63G |813        |248       |81.65        |
> |Step(d.1)       |16 |1.0 |False|True |[2,2,2,2] |True |22.9M |4.02G |1336 $\dagger$|405 $\dagger$|81.05 $\ddagger$|
> |Step(d.2)       |16 |1.0 |False|True |[2,2,5,2] |True |28.2M |5.18G |1137 $\dagger$|348 $\dagger$|82.24 $\ddagger$|
> |Step(e)         |16 |1.0 |False|False|[2,2,5,2] |True |26.2M |4.86G |1179       |360       |82.17        |
> |Step(e.1)       |16 |1.0 |True |False|[2,2,5,2] |True |26.3M |4.87G |1164       |358       |82.31        |
> |Step(e.2)       |1  |1.0 |True |False|[2,2,5,2] |True |25.6M |3.98G |1942       |647       |81.87        |
> |Step(f)         |1  |2.0 |True |False|[2,2,5,2] |True |30.7M |4.86G |1340       |464       |82.49        |
> |Step(g)         |1  |1.0 |True |False|[2,2,8,2] |True |30.2M |4.91G |1686       |571       |82.60        |
>
> Details of accelerating VMamba. $\dagger$ and $\ddagger$ indicate the value is obtained from `[SERVER 1]` and `[SERVER 2]`, respectively. All other experiments are conducted on `[SERVER 0]`.
>
>
> ### **Clarification of the Statement**
>
> There is a typo in the mentioned statement, and the correct version is "such modification prevents the weights from being input-dependent, resulting in a limited capacity for capturing contextual information" (i.e., change from "input-independent" to "input-dependent"). We will fix this typo and conduct thorough proofreading to prevent further errors in the revised version.
>
> **Detailed Explanation of the Referred Statement.** S4ND [2] extends S4 [1] to higher-dimensional contexts through a straightforward outer product, with the essential condition being that the SSM in S4 is implemented using 'accelerated convolution'. Specifically, S4 utilizes a global convolutional operation to compute the output of the SSM, denoted as $\mathbf{y}$, given the input data $\mathbf{u}$ and the kernel function $\mathbf{K} = \mathbf{C}e^{\mathbf{A\Delta}}\mathbf{B}$.
>
> Efficient computation is achieved if $\mathbf{A}$ has an 'Normal Plus Low-Rank' (NPLR) form and $\Delta$ is constant, enabling the low-rank approximation of $\mathbf{K}$ in the spectral domain, allowing the convolution to be efficiently computed with Fast Fourier Transform (FFT) and Inverse Fast Fourier Transform (IFFT). Conversely, if $\Delta$ is input-dependent or context-aware, the kernel function will no longer maintain a low-rank form in the spectral domain, leading to a substantial increase in the convolution computation time.
>
> The efficacy of a recurrent model is significantly limited by its capacity to effectively compress context [3]. By leveraging the task of selective copying and employing Induction heads, Mamba [3] illustrates that LTI models lack content awareness. Consequently, it is concluded that a fundamental principle in developing sequence models is selectivity: the context-aware capability to emphasize or disregard specific inputs within a sequential state.
>
> ### **Repetitive Reference Entries**
>
> We will address the issues mentioned in the comment and conduct thorough proofreading to prevent further errors in the revised version.

---

> ### Comment · Reviewer_ZfkW · 2024-08-12
> **Final Rating**
>
> Thanks to the authors for providing a thorough and solid response to all my concerns. Based on all the reviewers' comments and the rebuttal, I am happy to keep my rating as a Strong Accept (8).

---

### Official Review · Reviewer_dvTH · 2024-07-13

**Soundness:** 3
**Presentation:** 3
**Contribution:** 3
**Rating:** 7
**Confidence:** 3

**Summary:**

### Summary

This paper proposes VMamba, which adopts the recently proposed selective linear state space model, Mamba, in the domain of computer vision. The paper evaluates variants of VMamba on tasks such as image classification, object detection, and semantic segmentation. To improve performance and efficiency, VMamba incorporates several architectural and implementation enhancements.

---

post-rebuttal: score 6 -> 7

**Strengths:**

### Strengths

- The writing is simple and clear, quite accessible to readers.
- After implementing enhancements, VMamba achieves good performance - computational efficient and quantitatively well-performed.
- Additional analysis such as the effective receptive field and relationship between attention and the updates in state space are insightful.

**Weaknesses:**

### Weaknesses

- As an architecture exploration paper I don’t see many weaknesses.

**Questions:**

### Questions

1. Is positional embedding used when encoding the patches? Apologize if this is already state somewhere in the paper.
2. If I understand correctly, Figure 3 for section 4.3 shows that performance improved with smaller d_state and expand ratio. This is quite surprising since one might expect degrading performance when network capacity is reduced. Could you provide any insights into this phenomenon?

**Limitations:**

Yes, the author adequately addressed the limitations.

---

> ### Author Rebuttal · Authors · 2024-08-06
>
> # Response to Reviewer dvTH
>
> We thank the reviewer for the constructive comments and are glad they appreciate the performance of VMamba. Below, we clarify the reviewer’s concerns regarding the detailed structure and the influence of hyper-parameters on VMamba.
>
> ### **Usage of Positional Embedding**
>
> To clarify, VMamba does not use positional embedding. Sorry for any confusion caused, and we will make this clear in Section 4.1 Network Architecture (lines 129-136) of the revised version as follows:
>
> "Subsequently, multiple network stages are employed to create hierarchical representations"
> $\rightarrow$
> "Without further incorporating positional embedding, multiple network stages are employed to create hierarchical representations."
>
> ### **Explanation of Performance Improvement**
>
> In step (e) shown in Figure 3 for Section 4.3, we manage to save parameters and FLOPs by reducing the expansion ratio and eliminating the entire multiplicative branch. This allows us to increase the number of layers from [2,2,2,2] to [2,2,5,2], resulting in the observed performance improvement. Similarly, in step (g), lowering the expansion ratio enables us to increase the depth of the model with additional layers. For step (f), the performance improvement is due to the larger expansion ratio and the addition of extra DWConv blocks. By using a smaller d\_state value, we keep parameters and FLOPs comparable. We will provide more details on these points in the revised version.
>
> **Influence of d\_State.** In Section H.3 of the Appendix, we explore the impact of adjusting the d\_state parameter on VMamba. Table 12 shows that increasing d\_state from 1 to 4 yields only marginal performance gains while significantly reducing throughput, indicating a substantial negative impact on VMamba's computational efficiency. To mitigate this, we propose lowering the ssm\_ratio parameter to reduce overall network complexity. We find the best performance at (d\_state=8, ssm\_ratio=1.5).
>
> **Influence of ssm\_ratio.** We also analyze VMamba's sensitivity to the ssm\_ratio parameter, with results presented in Table 13 of Appendix H.4. The results clearly indicate that lowering the ssm\_ratio significantly reduces performance but also greatly increases the inference speed. On the other hand, adding more layers boosts performance but also decelerates the model.

---

> > ### Comment · Reviewer_dvTH · 2024-08-12
> >
> > Thank you for the rebuttal. I will raise my score, this is a good paper.

---

### Author Rebuttal · Authors · 2024-08-06

# Response to all

We thank the reviewers for their thoughtful reviews and constructive suggestions. We’re glad that the reviewers recognized the innovation and influence of the proposed 2D-Selective-Scan (SS2D) module, as well as the extensive experiments and thorough analysis supporting VMamba. In the following, we provide a shared response to common concerns raised by the reviewers, and also include a PDF file (referred to as the `attachment`) with additional experimental results to support our discussion. Additional results are included in the `attachment` (figures) as well as in the separate responses to each reviewer (tables).

### **Ablation Study on Hyper-parameters**

All reviewers have raised concerns regarding the influence of hyper-parameters. Due to the mismatch between our limited computational resources and the extensive range of design choices, we did not initially conduct a comprehensive ablation study on all hyper-parameters, focusing instead on a subset included in the appendix. As suggested by the reviewers, we have now conducted additional experiments on this topic.

### **Comparison with SSM-based Models**

Another focus of the reviewers is the need for a more in-depth comparison between VMamba and SSM-based models, such as S4ND [2] and Vim [4]. We recognize the importance of these comparisons and have conducted additional experiments as suggested. The results include comparisons of FLOPs, visualizations of the Effective Receptive Fields (ERFs), and analyses of the changes in performance (both effectiveness and efficiency) with increasing input resolution.

### **Statement on Experiment Platforms**

Please note that there are slight differences between the platforms we used for the original study and this rebuttal.

| Usage | CPU | GPU | Notation |
|:--:|:--:|:--:|:--:|
|Original Work|AMD EPYC 7542| 8 $\times$ Tesla A100 GPU|`[SERVER 0]`|
|Rebuttal (Testing)|Intel Xeon Platinum 8358|Tesla A800 GPUs|`[SERVER 1]`|
|Rebuttal (Training)|Intel Xeon Platinum 8480C|8 $\times$ Tesla H100 GPU|`[SERVER 2]`|

We investigate the influence of computational platforms on evaluation results as follows. For `[SERVER 0]` and `[SERVER 1]`, we test the generalizability to inputs with increased spatial resolutions, and the results are shown in the following table. Both training and inference throughput values are measured with a batch size of $32$ using PyTorch 2.2. The training throughput calculations include only the model forward pass, loss forward pass, and backward pass.

| Model | Image Size | Params (M) | FLOPs (G) | [SERVER 0] TP.  (img/s) |  [SERVER 0]  Train TP. (img/s)| [SERVER 1] TP.  (img/s) |  [SERVER 1]  Train TP. (img/s)|
|:--:|:--:|:--:|:--:|:--:|:--:|:--:|:--:|
|VMamba-Tiny|$224^2$ |30M|4.91G |1490|418|1463|453|
|VMamba-Tiny|$288^2$|30M|8.11G |947 |303|952 |305|
|VMamba-Tiny|$384^2$|30M|14.41G|566 |187|563 |187|
|VMamba-Tiny|$512^2$|30M|25.63G|340 |121|339 |120|
|VMamba-Tiny|$640^2$|30M|40.04G|214 |75 |216 |75 |
|VMamba-Tiny|$768^2$|30M|57.66G|149 |53 |149 |53 |

We also compare the differences between VMamba-T trained on `[SERVER 0]` and `[SERVER 2]` in the following table.

| Model | Params (M) | FLOPs (G) | LR | Top 1. (\%) |
|:--:|:--:|:--:|:--:|:--:|
|VMamba-Tiny [SERVER 0]|30M|4.91G|1e-3|82.60|
|VMamba-Tiny [SERVER 2]|30M|4.91G|1e-3|82.62|

According to the results shown in the above two tables, there is only a subtle difference between the results obtained on `[SERVER 0]` and `[SERVER 1]`/`[SERVER 2]`. Therefore, we disregard the influence of computational platforms and will include results obtained with consistent machines in the revised version.

### **Citations:**
    [1] Albert Gu, Karan Goel, and Christopher Re. Efficiently modeling long sequences with structured state spaces. In ICLR, 2021.

    [2] Eric Nguyen, Karan Goel, Albert Gu, Gordon Downs, Preey Shah, Tri Dao, Stephen Baccus, and Christopher Ré. S4nd: Modeling images and videos as multidimensional signals with state spaces. NeurIPS, 35:2846–2861, 2022.

    [3] Albert Gu and Tri Dao. Mamba: Linear-time sequence modeling with selective state spaces. arXiv preprint arXiv:2312.00752, 2023.

    [4] Lianghui Zhu, Bencheng Liao, Qian Zhang, Xinlong Wang, Wenyu Liu, and Xinggang Wang. Vision mamba: Efficient visual representation learning with bidirectional state space model. In ICML, 2024.

    [5] Ze Liu, Yutong Lin, Yue Cao, Han Hu, Yixuan Wei, Zheng Zhang, Stephen Lin, and Baining Guo. Swin transformer: Hierarchical vision transformer using shifted windows. In ICCV, pages 10012–10022, 2021.

---

### Decision · Program_Chairs · 2024-09-25

**Decision:**

Accept (spotlight)

**Comment:**

This paper introduces VMamba, a new vision backbone model inspired by the Mamba state-space model, designed for efficient visual representation learning with linear computational complexity. The key innovation is the VSS block, which incorporates a 2D-Selective-Scan (SS2D) module, adapting the 1D selective scan of Mamba for 2D image data. Experiments show that VMamba outperforms existing models in image classification, object detection, instance segmentation, and semantic segmentation, demonstrating superior accuracy, scalability, and adaptability. All reviewers are satisfied with the authors' responses.